# CellCLIP – Learning Perturbation Effects in Cell Painting via Text-Guided Contrastive Learning

**Mingyu Lu**,[*] **Ethan Weinberger**,[*] **Chanwoo Kim, Su-In Lee**
Paul G. Allen School of Computer Science & Engineering
University of Washington
{mingyulu,ewein,chanwkim,suinlee}@cs.washington.edu

## Abstract

High-content screening (HCS) assays based on high-throughput microscopy techniques such as Cell Painting have enabled the interrogation of cells' morphological responses to perturbations at an unprecedented scale. The collection of such data promises to facilitate a better understanding of the relationships between different perturbations and their effects on cellular state. Towards achieving this goal, recent advances in cross-modal contrastive learning could, in theory, be leveraged to learn a unified latent space that aligns perturbations with their corresponding morphological effects. However, the application of such methods to HCS data is not straightforward due to substantial differences in the semantics of Cell Painting images compared to natural images, and the difficulty of representing different classes of perturbations (e.g. small molecule vs CRISPR gene knockout) in a single latent space. In response to these challenges, here we introduce CellCLIP, a cross-modal contrastive learning framework for HCS data. CellCLIP leverages pre-trained image encoders coupled with a novel channel encoding scheme to better capture relationships between different microscopy channels in image embeddings, along with natural language encoders for representing perturbations. Our framework outperforms current open-source models, demonstrating the best performance in both cross-modal retrieval and biologically meaningful downstream tasks while also achieving significant reductions in computation time. Code for our reproducing our experiments is available at `https://github.com/suinleelab/CellCLIP`.

## 1   Introduction

A grand challenge in cellular biology is understanding the impacts of different perturbations, such as exposure to chemical compounds or gene knockouts, on cellular function. In pursuit of this goal, a number of high-content screening (HCS) techniques have been developed that combine image-based deep phenotyping with high-throughput perturbations [20, 30]. For example, the Cell Painting protocol [4] stains cells with six fluorescent dyes highlighting distinct cellular components (e.g. Hoescht staining for nuclear DNA), which are then imaged in five channels via automated microscopy (Figure 1a). Cell Painting is now routinely combined with other biotechnological advances to measure cells' response to thousands of chemical and/or genetic perturbations in parallel at low cost [40, 36].

Despite the promise of this data, extracting meaningful representations of cells' underlying states from their images presents formidable challenges. Traditional HCS data analyses use custom software tools to extract domain-expert-crafted morphological features (e.g. nucleus size) from each image, with the resulting collection of features referred to as a cell's *profile* [6, 8]. More recent works have found that self-supervised deep learning methods based on DINO [7, 33] or masked autoencoder (MAE; [21])

---

[*]Equal contribution.

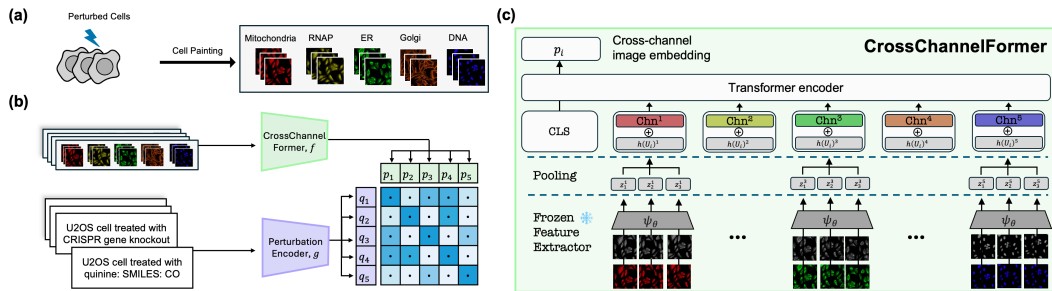

Figure 1: Overview of CellCLIP. (**a**) Cells' responses to a perturbation are measured by the Cell Painting assay, which captures a set of distinct cellular components in five imaging channels. (**b**) CellCLIP aligns perturbation embeddings generated from natural language descriptions with embeddings of corresponding images produced by a new CrossChannelFormer architecture designed to account for the idiosyncrasies of Cell Painting. (**c**) For a perturbation $i$, CrossChannelFormer takes as input images of all cells receiving perturbation $i$ and pools them into a single embedding $p_i$ that takes into account the relationship between information contained in the different Cell Painting channels.

architectures can capture more subtle changes in morphology compared to human-designed features, resulting in profiles that better reflect known relationships between perturbations.

In order to more explicitly capture the relationships between perturbations' effects on cellular state, recent work has applied cross-modal contrastive learning (CL) techniques [35] to learn aligned representations of both perturbation labels and Cell Painting images. In particular, recent work has applied CL to this task by treating images and their corresponding perturbation labels as paired samples from different modalities [16, 38]. Post-training, such methods can be applied to systematically identify perturbations with similar effects based on relative proximity in the perturbation encoder's representation space. Moreover, novel perturbations can be input to the perturbation encoder to predict their morphological effects based on distance to previously seen perturbations.

Despite their promise, previous CL methods for Cell Painting come with significant drawbacks. First, unlike natural images, where the standard RGB channels all capture semantically related phenomena, microscopy images are comprised of a larger number of channels each capturing semantically independent information. Second, Cell Painting image-perturbation pairs exhibit a "many-to-one" structure where multiple images are associated with the same perturbation label. Standard CL, however, treats all image pairs from different instances as negatives, even if they share the same label, leading to a substantial number of *false negatives* where perceptually similar images are incorrectly pushed apart in the embedding space [5, 45]. Furthermore, not all Cell Painting images faithfully represent the intended perturbation effect, as some perturbations may be unsuccessful due to technical issues (e.g., poor guide efficiency with CRISPR-mediated gene knockouts), effectively resulting in false positive pairs. Yet, existing CL methods for HCS data fail to account for these properties [38] or are not openly available [16]. Finally, previous works [16, 38] have exclusively focused on chemical perturbations, relying on graph representation learning techniques applied to chemical structures to obtain perturbation representations. Thus, these methods cannot be applied to other classes of perturbations (e.g. CRISPR gene knockouts), and it is unclear how to represent perturbations from different classes within a unified input space for CL.

To address these issues, here we propose CellCLIP (Figure 1b), a CL framework designed for the unique challenges in Cell Painting perturbation data. Our framework employs off-the-shelf pretrained vision models alongside a novel channel-aware encoding scheme (Figure 1c) to facilitate reasoning across the different sources of information captured in Cell Painting channels; in addition, our image encoding architecture incorporates techniques from the multiple-instance-learning literature [24] to account for the many-to-one nature of Cell Painting perturbation-image pairs. To represent images' corresponding perturbations, CellCLIP employs natural language encoders, enabling application to multiple classes of perturbations with a single model. We benchmarked CellCLIP on profile-perturbation retrieval for unseen compounds [4] and its ability to recover known biological relationships between perturbations [10, 28]. We find that CellCLIP achieves strong performance compared to the previous state-of-the-art while taking a fraction of the time to train. Our method also shows promising results in cross-perturbation class matching, thereby providing a scalable and effective solution for biological discovery.

## 2  Background

**Representation Learning for Cell Painting**   Self-supervised learning (SSL) deep learning techniques have been successfully applied to experimental microscopy data, with recent studies demonstrating their ability to capture intricate details of cellular morphology [40, 13] better than hand-crafted features as implemented in standard software packages like CellProfiler [8]. Initial applications of SSL methods to Cell Painting relied on architectures designed for natural RGB images, where information across channels exhibits strong correlations [2]. On the other hand, Cell Painting channels each capture distinct biological structures (e.g. actin via phalloidin, mitochondria via MitoProbe, etc.). To account for this, recent works [2, 27] have proposed so-called channel-agnostic vision transformers (CA-ViTs), that use separate tokens for each channel in a spatial patch rather than aggregating information across all channels. However, CA-ViTs substantially increase computational costs due to an increased number of tokens, since each sub-patch corresponds to a token.[2]

**Contrastive Learning**   Cross-modal contrastive learning methods such as CLIP [35] learn aligned latent representations across multiple modalities (e.g., text and image). Such methods optimize a symmetric contrastive loss to maximize the similarity between correct pairs while minimizing it for incorrect ones. Specifically, given a batch of $N$ cross-modal pairs $\{(u_i, v_i)\}_{i=1}^{N}$ where $u \in \mathcal{U}$ and $v \in \mathcal{V}$ (e.g. text and images), let $f_\theta$ and $g_\phi$ be encoders that map inputs to embeddings $p_i = f_\theta(u_i)$ and $q_j = g_\phi(v_j)$, respectively. The CLIP loss consists of cross entropy objectives encouraging alignment from $\mathcal{V} \to \mathcal{U}$ and $\mathcal{U} \to \mathcal{V}$, with $\tau$ as a learnable temperature parameter:

$$\mathcal{L}_{\text{CLIP}} = \frac{1}{N} \sum_{i=1}^{N} \left[ \underbrace{-\log \frac{\exp(\langle p_i \cdot q_i \rangle / \tau)}{\sum_{j=1}^{N} \exp(\langle p_i \cdot q_j \rangle / \tau)}}_{\mathcal{U} \to \mathcal{V}} + \underbrace{-\log \frac{\exp(\langle p_i \cdot q_i \rangle / \tau)}{\sum_{j=1}^{N} \exp(\langle p_j \cdot q_i \rangle / \tau)}}_{\mathcal{V} \to \mathcal{U}} \right] \quad (1)$$

While CL excels in cross-modality alignment, several challenges remain in its application to Cell Painting perturbation data. First, while a thorough body of work exists studying encoding strategies for natural images and natural language [39, 46, 1], encoding strategies specifically designed for Cell Painting images and cellular perturbations remain largely underexplored. Second, each perturbation is typically associated with multiple images, so naively attempting to align individual Cell Painting images with perturbation labels may introduce a substantial number of false negatives. Third, perturbations in Cell Painting experiments can span a variety of classes, including chemical perturbations, gene knockouts, and open reading frames (ORFs). Yet, previous contrastive learning approaches for Cell Painting, such as CLOOME [38] and MolPhenix [16] have exclusively focused on chemical perturbations, limiting their applicability. Finally, model weights for some recent state-of-the-art methods (e.g. [16]) are not openly available, and retraining these models from scratch is infeasible due to the use of large closed-source datasets and significant GPU resources. Thus, there exists a need for high-performance open-source alternatives.

## 3  Methods

Here we present CellCLIP in detail. We first present CrossChannelFormer, our image encoding scheme that facilitates CL with Cell Painting data (Section 3.1); due to the idiosyncracies of Cell Painting perturbation datasets, this encoding must be done with care, and we cannot naively reuse strategies for CL with natural images. We then present our approach for encoding perturbation labels (Section 3.2) and outline our training strategy (Section 3.3).

### 3.1  CrossChannelFormer

Our CrossChannelFormer architecture for image encoding (Figure 1c) consists of three steps that allow us to effectively account for the nuances of Cell Painting perturbation data without introducing prohibitive computational costs. We describe these steps below.

**Step 1: Cell profiles from natural image foundation models.**   Recent works have developed image foundation models trained on natural images, such as DINOv2 [33], which have demonstrated strong

---

[2]For an image of size $h \times w$ with $c$ channels, CA-ViTs produce $\frac{h \times w}{p^2} \times c$ tokens assuming a patch size $p \times p$.

capabilities in capturing global structural image features. Rather than training new models from scratch, we choose to adopt these pretrained models as image encoders in the CellCLIP framework. However, unlike natural images with the standard set of RGB channels, Cell Painting images contain a variable number of channels corresponding to the specific stains used in an experiment. To work around this difference and enable models trained on natural images to be applied to Cell Painting data, we treat each Cell Painting channel as an independent grayscale image and extract embeddings separately. Formally, for each perturbation $i$ we denote the collection of Cell Painting images corresponding to that perturbation as $U_i = \{u_k\}_{k=1}^{N_i}$, where $u_k \in \mathbb{R}^{C \times H \times W}$ denotes an individual image. For each image $u_k$, we may apply a feature extractor $\psi_\theta$ that maps individual channels $u_k^c$ to an embedding $z_k^c = \psi_\theta(u_k^c) \in \mathbb{R}^d$. This produces a channel-wise embedding matrix

$$z_k = [z_k^1, z_k^2, \ldots, z_k^C]^T \in \mathbb{R}^{C \times d}, \tag{2}$$

that we employ as an image's profile. In our experiments, we used a frozen, pretrained DINOv2 (giant) model for our feature extractor $\psi$, and explore the impact of different choices for $\psi$ in Appendix A.1.

**Step 2: Pooling profiles within perturbations.** The standard CLIP model aligns pairs of natural images and corresponding text annotations. However, the presence of both false negative (i.e., images sharing the sample perturbation label) and false positive pairs (i.e., images resembling controls) may lead to subpar results when attempting to align individual Cell Painting image profiles and corresponding perturbation annotations. To accommodate this issue, instead of aligning pairs of perturbation labels and individual images, we instead align perturbation labels with an aggregated summary of all images that received that perturbation. Specifically, for the set of images $U_i$ corresponding to perturbation $i$, we define their corresponding pooled profile for channel $c$ as

$$\mu(U_i)^c = \mathcal{S}\left(\{z_k^c \mid k = 1, \ldots, N_i\}\right) \in \mathbb{R}^d, \tag{3}$$

where $\mathcal{S}(\cdot)$ denotes a permutation-invariant transformation function applied channel-wise (e.g., a mean computed separately for each channel). For our experiments we use the gated attention pooling operator of Ilse et al. [24] for $\mathcal{S}$, and we explore the impact of different choices for $\mathcal{S}$ in Appendix A.2. In addition to improving performance on downstream tasks, per-perturbation pooling also improves computational efficiency: instead of computing a contrastive loss for each of the $N_i$ individual profiles associated with a perturbation $i$, we compute a single pooled representation. This reduces the number of positive pairs per perturbation from $N_i$ to one, leading to an approximate $N_i$-fold speedup in training time compared to instance-level supervision.

**Step 3: Reasoning across information from different channels.** Beyond just a different number of channels, the relationships between Cell Painting image channels exhibit substantial differences compared to those between natural image channels. In particular, while natural image channels share a significant amount of information, Cell Painting channels correspond to distinct biological stains, each highlighting independent cellular structures. Thus, to effectively learn meaningful embeddings of Cell Painting images, it is necessary to explicitly reason between information in different channels [2], and naively reusing existing vision encoders designed for RGB images as done by CLOOME [38] may produce subpar performance. While previously proposed CA-ViT architectures [2] do enable cross-channel reasoning, as discussed previously this ability comes at the cost of substantially increased computational overhead due to the larger number of tokens required.

Thus, to enable cross-channel reasoning while minimizing computational costs, we introduce Cross-ChannelFormer, a specialized encoder architecture for CellCLIP. Unlike the standard Vision Transformer (ViT; [1]), where each input token represents a multi-channel image patch, CrossChannelFormer takes pooled profiles $\mu(U_i)$ (Equation (3)) that capture the *global* cellular features associated with specific stains. We then introduce a set of learnable channel embeddings, $[\text{chn}^1, \ldots, \text{chn}^C]$, where each $\text{chn}^c \in \mathbb{R}^d$ encodes information unique to its respective channel. Finally, we prepend a learnable classifier token $\text{cls} \in \mathbb{R}^d$ to the sequence, which aggregates global image features across all channels. The resulting input sequence to the transformer is:

$$[\text{cls}, \mu(U_i)^1 + \text{chn}^1, \mu(U_i)^2 + \text{chn}^2, \ldots, \mu(U_i)^C + \text{chn}^C], \tag{4}$$

where $\mu(U_i)^c$ corresponds to the $c$-th channel of $\mu(U_i)$. This design is inspired by Bao et al. [2] but differs in that we reason over the global per-channel information captured in $\mu(U_i)$ rather than local per-channel spatial patches. This results in a substantially reduced computational burden compared to CA-ViTs [2, 27] as CrossChannelFormer requires only $C + 1$ tokens for each perturbation.

Following the original ViT, we feed the above sequence into a Transformer encoder. The Transformer encoder consists of alternating layers of multi-head self-attention and MLP blocks, with layer normalization applied before each block and residual connections established after each block. The final layer representation of the CLS token serves as the projection in the latent space.

Altogether, our proposed CrossChannelFormer image encoding strategy offers three main advantages over previous approaches. First, our approach allows us to reuse off-the-shelf vision encoders pretrained on natural image data, which are far more plentiful than specialized models pretrained on Cell Painting images. Second, by aligning pooled image profiles with perturbation labels instead of individual images, we effectively resolve the many-to-one nature of Cell Painting perturbation data. Finally, our CrossChannelFormer technique captures relationships between Cell Painting channels with significantly better computational efficiency.

## 3.2 Perturbation Encoding via Natural Language

Previous contrastive learning methods for Cell Painting rely on perturbation-class-specific encoders to represent perturbation treatments. For instance, CLOOME [38] encodes chemical compounds by passing Morgan fingerprints [32] through a simple multilayer perceptron (MLP), and Molphenix relies on graphical neural networks (GNN) [16]. This setup is not ideal, as different perturbation types (e.g., chemical compound vs gene knockouts) require distinct encoder networks, making it challenging to incorporate data from multiple perturbation types into the contrastive learning process.

To address this, we adopt a simple approach—representing each perturbation using *text*. Since most perturbations and their associated metadata can be effectively captured through textual descriptions [37, 11], text serves as an efficient intermediate modality for generalizing across multiple perturbation types. We construct a corresponding text prompt that encodes information on cell types and perturbation-specific details. For example, to encode the chemical compound butyric acid, a drug affecting cell growth, we use the prompt:

> *"A cell painting image of U2OS cells treated with butyric acid, SMILES: CCCC(O)=O."*

Similarly, for a CRISPR perturbation, the prompt is structured as:

> *"A cell painting image of U2OS cells treated with CRISPR, targeting genes: AP2S1."*

By representing perturbations as text prompts, our approach facilitates encoding arbitrary perturbations from different classes, simplifying training across diverse perturbation types. It can also potentially integrate relevant textual metadata, enhancing perturbation retrieval across experiments.

For our experiments, we use a pretrained BERT model [12] which uses the WordPiece tokenizer [46] and supports 512 tokens. The prompt template used is provided in Appendix B.

## 3.3 CellCLIP Training Objective

In standard contrastive learning, the objective is to align matched data pairs by minimizing the loss in Equation (1). In Cell Painting, in addition to cross-modal retrieval [16, 38], many *intra-modal* downstream tasks, such as matching related perturbations [10, 25, 28], are also important. However, maintaining adequate performance on intra-modal tasks is non-trivial; for example, in the context of Cell Painting perturbation screens, many distinct perturbations induce highly similar morphological profiles, i.e., $f_{\theta^*}(\mu(U_i)) \approx f_{\theta^*}(\mu(U_j))$ for $i \neq j$, where $f_{\theta^*}$ is an oracle profile encoder. In this setting, treating all non-matching pairs as hard negatives can degrade retrieval performance by encouraging too much separation between biologically similar perturbations.

To address this, we adopt the Continuously Weighted Contrastive Loss (CWCL) [41] in CellCLIP. CWCL replaces binary labels with continuous similarity-based weights (Figure 1b), applying a soft labeling scheme within the cross-entropy loss. The loss helps training cross-modal models by ensuring that similarities picked up by a strong pretrained unimodal embedding model are maintained in the cross-modal representation space. In our case, we apply the CWCL loss by computing target similarities between per-perturbation pooled profiles so that morphologically similar profiles remain close in CellCLIP's embedding space. For the profile-to-perturbation direction, $\mathcal{U} \rightarrow \mathcal{V}$, our adapted CWCL objective is given by

$$\mathcal{L}_{\text{CWCL},\mathcal{U}\rightarrow\mathcal{V}} = \frac{1}{N} \sum_{i=1}^{N} \frac{1}{\sum_{j\in[N]} w_{ij}^{\mathcal{U}}} \left[ \sum_{j=1}^{N} w_{ij}^{\mathcal{U}} \cdot \log \frac{\exp(\langle p_i \cdot q_j \rangle/\tau)}{\sum_{k=1}^{N} \exp(\langle p_i \cdot q_k \rangle/\tau)} \right]. \tag{5}$$

Here $p_i$ represents the output of CrossChannelFormer applied to images corresponding to perturbation $i$, $q_i$ corresponds to our encoding of the natural language description of perturbation $i$, and $w_{ij}^{\mathcal{U}}$ denotes the similarity between the pooled profiles $\mu(U_i)$ and $\mu(U_j)$, used to reweight the alignment from modality $\mathcal{U}$ to modality $\mathcal{V}$. We compute $w_{ij}^{\mathcal{U}}$ as the average channel-wise cosine similarity between pooled profiles.[3] In Appendix A.4 we provide results illustrating the benefits of our choice over other potential contrastive losses.

For the perturbation-to-profile direction, $\mathcal{V} \rightarrow \mathcal{U}$, we apply the standard CLIP loss (Equation (1)) as we found computing similarities either within the input space $\mathcal{V}$ (i.e., discrete tokens) or in the projected space $\mathcal{Q}$ (i.e., BERT embeddings) did not provide meaningful signals for soft-label supervision. Thus, the final training objective is the sum of the two directional losses:

$$\mathcal{L}_{\text{total}} = \mathcal{L}_{\text{CWCL}, \mathcal{U} \rightarrow \mathcal{V}} + \mathcal{L}_{\text{CLIP}, \mathcal{V} \rightarrow \mathcal{U}}. \tag{6}$$

## 4 Experimental Setup

In this section, we describe the datasets and tasks used to evaluate our framework.

### 4.1 Cross-Modality Retrieval Between Profiles and Perturbations

We first benchmarked CellCLIP's performance by assessing its ability to retrieve test set perturbations given corresponding Cell Painting image profiles treated with each perturbation as done in prior works [16, 35, 38]. That is, for a given model we compute perturbation embeddings $\mathcal{Q}$ along with corresponding pooled Cell Painting profile embeddings $\mathcal{P}$ in the shared latent space. Given the per-perturbation pooled Cell Painting profile embeddings, we then compute cosine similarities with all perturbations' embeddings in the test set and retrieve the top-$k$ most similar perturbations; ideally, the image profiles' true perturbation should be contained in this nearest neighbors set. Our evaluation metric, Recall@$k$ (R@$k$), measures whether the correct perturbation appears in the top-$k$ retrieved results, with $k = 1, 5, 10$. We denote this task as *profile-to-perturbation* retrieval.

Swapping the roles of perturbations and Cell Painting profiles, we may similarly evaluate *perturbation-to-profile* retrieval, where, given a perturbation embedding, we compute similarities with pooled Cell Painting profile embeddings. For this task we again use Recall@$k$ for evaluation.

### 4.2 Intra-Modality Retrieval on Biologically Meaningful Tasks

To further assess if CellCLIP's image encoder captures biologically meaningful signals, we evaluated its profile embeddings $\mathcal{P}$ on the following three *intra-modal* retrieval tasks proposed in Chandrasekaran et al. [10] and Kraus et al. [28]. For each task, we assess if the embeddings corresponding to cells receiving a given perturbation are close to those for other biologically related perturbations.

- **Replicate detection** measuring how well replicates across batches of a given perturbation can be distinguished from negative controls. In other words, profiles treated with the same perturbation—but collected in separate batches—should be close in embedding space.

- **Sister perturbation matching** assesses whether cells treated with "sister" perturbations targeting the same genes, which should thus induce similar morphological changes, are close in embedding space. This includes comparisons within the same perturbation class (e.g., compound-compound) and across different classes (e.g., CRISPR-compound).

- **Zero-shot gene-gene relationship recovery** evaluates models' generalization by assessing whether their profile embeddings for cells receiving gene knockout perturbations in a previously unseen dataset reflect known relationships between genes. Methods that construct accurate relational maps may be more likely to uncover novel biological interactions.

For our evaluations on the above tasks, we compute batch-effect-corrected pooled embeddings $\tilde{p}_i$ using matched negative controls as in Celik et al. [9]. For a given condition $i$, we compute the cosine similarity between $\tilde{p}_i$ and embeddings for other conditions $\tilde{p}_j$ and then rank them to perform retrieval.

---

[3]Specifically, $w_{ij}^{\mathcal{U}} = \sum_{c=1}^{C} \frac{\langle \mu(U_i)^c, \mu(U_j)^c \rangle}{2C} + 0.5$, ensuring $w_{ij}^{\mathcal{U}} \in [0, 1]$.

Table 1: Benchmarking CellCLIP and baseline methods on perturbation-to-profile and profile-to-perturbation retrieval performance for unseen molecules from Bray et al. [4]. We report mean Recall@1, @5, and @10 $\pm$ standard deviation across random seeds for both tasks. Higher recall corresponds to better performance. Best results are shown in **bold**.

| Model | Perturb-to-profile (%) ↑ | | | Profile-to-perturb (%) ↑ | | |
|---|---|---|---|---|---|---|
| | R@1 | R@5 | R@10 | R@1 | R@5 | R@10 |
| CLOOME | $0.27 \pm 0.20$ | $1.25 \pm 0.42$ | $2.46 \pm 0.56$ | $0.07 \pm 0.09$ | $1.44 \pm 0.48$ | $2.56 \pm 0.89$ |
| CLOOME‡ | $0.45 \pm 0.13$ | $1.60 \pm 0.12$ | $3.04 \pm 0.12$ | $0.48 \pm 0.16$ | $1.79 \pm 0.13$ | $3.12 \pm 0.25$ |
| CLOOME‡ (CLOOB) | $0.37 \pm 0.10$ | $1.56 \pm 0.10$ | $2.75 \pm 0.02$ | $0.20 \pm 0.03$ | $1.76 \pm 0.10$ | $3.13 \pm 0.17$ |
| MolPhenix* (S2L) | $0.28 \pm 0.11$ | $1.56 \pm 0.19$ | $3.00 \pm 0.26$ | $0.42 \pm 0.08$ | $1.54 \pm 0.03$ | $3.01 \pm 0.20$ |
| MolPhenix* (SigCLIP) | $0.56 \pm 0.12$ | $2.78 \pm 0.23$ | $4.30 \pm 0.16$ | $0.66 \pm 0.22$ | $2.55 \pm 0.05$ | $4.34 \pm 0.24$ |
| CellCLIP (Ours) | $\mathbf{1.18 \pm 0.20}$ | $\mathbf{4.49 \pm 0.06}$ | $\mathbf{7.37 \pm 0.20}$ | $\mathbf{1.25 \pm 0.10}$ | $\mathbf{4.82 \pm 0.10}$ | $\mathbf{7.39 \pm 0.23}$ |

‡ Indicates that CLOOME uses the same image profile encoding procedure and architecture as MolPhenix*

Retrieval performance is evaluated against ground-truth labels via task-specific metrics used in previous work. For replicate detection and sister perturbation matching, we report mean average precision (mAP) as in Chandrasekaran et al. [10]. For biological relationship recovery, we follow Kraus et al. [28] and report recall over the top and bottom 5% of the similarity distribution (i.e., 10% recall). Further details regarding the computation of these metrics are provided in Appendix C.

### 4.3 Datasets

For **cross-modality retrieval**, following Sanchez-Fernandez et al. [38], we utilize a curated version of Bray et al. [4], comprising approximately 284,034 five-channel Cell Painting images corresponding to 10,578 small-molecule perturbations. We partitioned the dataset by perturbation into train, validation, and test sets with a 70/10/20 split, resulting in 2,115 unseen small molecules in the test set.

For **replicate detection & sister perturbation matching**, we employ CPJUMP1 [10], which features 186,925 eight-channel microscopy images. These images include three bright-field channels in addition to five Cell Painting dye channels. They are perturbed across 650 distinct perturbations, including compounds and genetic modifications such as CRISPR and ORF interventions. For each perturbation class, we applied a 70/10/20 split for training, validation, and testing.

For **zero-shot gene-gene relationship recovery**, we use RxRx3-core [28], a curated subset of RxRx3 [15], covering 736 gene knockouts. Gene-gene relationships are obtained from a set of public databases (Reactome, HuMAP, SIGNOR, StringDB, and CORUM; Table S.10) as in Kraus et al. [27].

Further details on datasets and preprocessing are provided in Appendix D.

### 4.4 Model training

For cross-modality retrieval evaluation on Bray et al. [4], CellCLIP was trained with 50 epochs using a batch size of 512 and an AdamW optimizer. The learning rate was set at $2 \times 10^{-4}$ with cosine annealing and restart. The temperature parameter, $\tau$, is initialized to 14.3. For replicate detection and sister perturbation matching in CPJUMP1 [10], we reused our model trained on the Bray et al. [4] dataset and fine-tuned it for another 50 epochs, using the same parameter settings as during their initial training. For gene relationship recovery using RxRx3-core, we perform this evaluation in a "zero-shot" manner and evaluate models trained on Bray et al. [4] without subsequently fine-tuning on RxRx3-core. More details about training CellCLIP and other baselines can be found in Appendix B.

## 5 Results

### 5.1 Cross-modality retrieval

We present our results for cross-modality retrieval tasks (i.e., perturbation-to-profile and profile-to-perturbation) in Table 1. For our benchmarking we compared CellCLIP against previous state-of-the-art CL methods for Cell Painting perturbation screens: CLOOME [38] and MolPhenix [16]. For CLOOME we used the same architecture with optimal hyperparameters as specified in Sanchez-

Table 2: Ablation studies of various vision and perturbation encoder combinations and retrieval performance in [4]. We report mean Recall@1, @5, and @10 ± standard deviation across random seeds for perturbation-to-profile and profile-to-perturbation retrieval tasks. △ indicates Morgan Fingerprint; □ indicates text prompt. Best results are shown in **bold**.

| Vision Encoder | Perturb. Encoder | Train Time (hr) | Perturb-to-profile (%) ↑ | | | Profile-to-perturb (%) ↑ | | |
|---|---|---|---|---|---|---|---|---|
| | | | R@1 | R@5 | R@10 | R@1 | R@5 | R@10 |
| ResNet-101 | △ + MLP | 49.9 | $0.27 \pm 0.20$ | $1.25 \pm 0.42$ | $2.46 \pm 0.56$ | $0.07 \pm 0.09$ | $1.44 \pm 0.48$ | $2.56 \pm 0.89$ |
| ResNet-101 | △ + MPNN++ | 45.2 | $0.20 \pm 0.06$ | $1.30 \pm 0.31$ | $3.03 \pm 0.21$ | $0.15 \pm 0.12$ | $1.50 \pm 0.41$ | $2.78 \pm 0.51$ |
| ResNet-101 | □ + BERT | 40.5 | $0.74 \pm 0.07$ | $2.67 \pm 0.09$ | $4.50 \pm 0.29$ | $\mathbf{1.10 \pm 0.46}$ | $3.95 \pm 0.49$ | $5.97 \pm 0.57$ |
| CA-MAE | □ + BERT | 25.6 | $0.22 \pm 0.07$ | $1.57 \pm 0.04$ | $2.80 \pm 0.05$ | $0.19 \pm 0.05$ | $1.29 \pm 0.10$ | $2.52 \pm 0.14$ |
| ChannelViT | □ + BERT | 29.5 | $0.78 \pm 0.15$ | $3.23 \pm 0.32$ | $5.15 \pm 0.52$ | $0.69 \pm 0.18$ | $2.92 \pm 0.39$ | $5.01 \pm 0.38$ |
| CrossChannelFormer[†] | □ + BERT | 12.2 | $\mathbf{1.16 \pm 0.12}$ | $3.98 \pm 0.33$ | $5.74 \pm 0.36$ | $1.05 \pm 0.08$ | $3.84 \pm 0.43$ | $5.90 \pm 0.28$ |
| CrossChannelFormer | □ + BERT | **1.81** | $1.18 \pm 0.20$ | $\mathbf{4.49 \pm 0.06}$ | $\mathbf{7.37 \pm 0.20}$ | $\mathbf{1.25 \pm 0.10}$ | $\mathbf{4.82 \pm 0.10}$ | $\mathbf{7.39 \pm 0.23}$ |

[†]: Trained without profile pooling.

Table 3: Retrieval performance of CellCLIP trained with various perturbation-related prompts. Recall@1, @5, and @10 ± standard deviation across random seeds for perturbation-to-profile and profile-to-perturbation retrieval tasks.

| Prompt modification | Perturb-to-profile (%) | | | Profile-to-perturb (%) | | |
|---|---|---|---|---|---|---|
| | R@1 | R@5 | R@10 | R@1 | R@5 | R@10 |
| Original | $1.18 \pm 0.20$ | $4.49 \pm 0.06$ | $\mathbf{7.37 \pm 0.20}$ | $1.25 \pm 0.10$ | $4.82 \pm 0.10$ | $\mathbf{7.39 \pm 0.23}$ |
| Without SMILES string | $0.13 \pm 0.05$ | $0.68 \pm 0.19$ | $1.26 \pm 0.14$ | $0.14 \pm 0.07$ | $0.63 \pm 0.17$ | $1.24 \pm 0.13$ |
| Without cell type information | $1.11 \pm 0.13$ | $4.02 \pm 0.25$ | $6.59 \pm 0.33$ | $1.10 \pm 0.23$ | $4.07 \pm 0.29$ | $6.36 \pm 0.32$ |
| Without drug name | $\mathbf{1.47 \pm 0.29}$ | $\mathbf{4.77 \pm 0.20}$ | $7.37 \pm 0.29$ | $\mathbf{1.61 \pm 0.26}$ | $\mathbf{4.89 \pm 0.19}$ | $7.21 \pm 0.27$ |
| With DrugBank description | $1.13 \pm 0.16$ | $4.26 \pm 0.43$ | $6.94 \pm 0.37$ | $1.16 \pm 0.17$ | $4.34 \pm 0.36$ | $6.71 \pm 0.26$ |

Fernandez et al. [38], along with subsequent variations of the model described in Fradkin et al. [16]. Because Fradkin et al. [16]'s implementation of MolPhenix relied on closed-source components trained on non-public datasets, compared against an approximation of their framework which we designate as MolPhenix*. We refer to Appendix E for further details.

Overall, we found that CellCLIP demonstrated substantially higher performance at both cross-modal retrieval tasks compared to baseline methods. To understand the source of performance gains in CellCLIP, we conducted a series of ablations to understand the contribution of each component in CellCLIP to retrieval performance. Specifically, starting with CLOOME's proposed encoding scheme (Appendix E), where individual images encoded using ResNet50 are aligned with chemical perturbations encoded using Morgan fingerprints combined with an MLP, we gradually replaced each of CLOOME's components with those of CellCLIP and assessed each change's impact on model performance (Table 2). We describe our findings from these experiments in detail below.

**Language can effectively represent perturbations.** We began by replacing CLOOME's chemical structure encoder, which feeds chemicals' Morgan fingerprints through a multi-layer perceptrion (MLP), with the natural language encoder used in CellCLIP while holding all other model components fixed. We found that this change alone yielded significant performance gains for both retrieval tasks. Notably, we found this result continued to hold even after replacing the generic MLP used in CLOOME with an MPNN++ network designed specifically for molecular property prediction [31].

**Effect of perturbation prompts.** We next investigated alternative prompting strategies to assess the contribution of individual components. To do so, we re-trained CellCLIP after removing specific elements of the prompt template (Section 3.2), such as cell type, drug name, SMILES string, one at a time (Table 3). We found that removing the SMILES string caused a substantial drop in performance, while removing cell type information produced a smaller degradation. In contrast, removing drug names had little effect, likely because the richer SMILES representation already captures most of the relevant information.

We also tested whether prompts could be enriched with external metadata by incorporating DrugBank descriptions [26] for compounds present in the database. However, this did not yield performance improvements over the original template. A likely reason is the limited coverage of DrugBank, which includes only about 10% of the compounds in Bray et al. [4], as many of them are experimental

Table 4: Benchmarking results for replicate detection and sister perturbation matching on CP-JUMP1 [10], and gene-gene relationship recovery with RxRx3-core [28]. Performance is evaluated using mean average precision (mAP) across perturbations for replicate detection and sister perturbation matching, and recall for relationship recovery. Higher values correspond to better performance.

| Method | Replicate Detection (mAP) ↑ | Sister Perturb. Matching (mAP) ↑ | | Gene-Gene Relationship Recovery (Recall) ↑ | | | | |
|---|---|---|---|---|---|---|---|---|
| | | Within Class | Across Class | CORUM | HuMAP | Reactome | SIGNOR | StringDB |
| *Cross-modal* | | | | | | | | |
| CLOOME | .575 | .245 | .026 | .597 | .679 | .327 | .309 | .510 |
| MolPhenix* | .531 | .222 | .011 | .539 | .599 | .330 | .297 | .476 |
| CellCLIP | **.663** | **.413** | **.043** | **.714** | **.778** | **.427** | **.388** | **.618** |
| *Unimodal (self-supervised)* | | | | | | | | |
| OpenPhenom-S/16 | .357 | .219 | .031 | .649 | .723 | .418 | .386 | .579 |
| *Unimodal (weakly supervised)* | | | | | | | | |
| ViT-L/16 | .513 | .283 | .032 | .681 | .758 | .388 | .380 | .587 |

molecules. Overall, these findings suggest that our original prompting strategy is already highly effective.

**Cross-channel reasoning improves retrieval performance.** We next investigated the impact of CrossChannelFormer's ability to reason across global Cell Painting channel information. To do so, we replaced CLOOME's ResNet image encoder with our CrossChannelFormer encoder. To isolate the effects of CrossChannelFormer's ability to reason across channels from the effects of per-perturbation pooling, in this experiment we removed the CrossChannelFormer's pooling function and trained the resulting model (denoted as CrossChannelFormer[†] in Table 2) on individual image-perturbation pairs.

We found that this change led to an additional increase in model performance. To understand how our approach compared to previous channel-aware vision encoding approaches, we ran this same experiment using a channel agnostic MAE (CA-MAE) and ViT (ChannelViT) as vision encoders. We found that CrossChannelFormer consistently outperformed these baselines on both tasks. This demonstrates that image embeddings extracted from models pretrained on natural images can be effectively leveraged with CrossChannelFormer. Additionally, our approach substantially reduced training time, achieving a 3.9 times speedup compared to CLOOME and a 2.2 times speedup compared to other channel-agnostic methods. This efficiency stems from CrossChannelFormer operating in a compact feature space and requiring only $C + 1$ tokens per instance.

**Pooling yields improved alignment and computational efficiency.** Finally, we evaluated the impact of the attention-based pooling operator [24] within CrossChannelFormer compared to instance-level training (i.e., no pooling). We found that including pooling resulted in yet another increase in model performance. In addition, by reducing the number of pairs for contrastive loss computation, pooling yielding a 6.7 times speedup in training time relative to instance-level training. We also explored the impact of varying our choice of pooling operator (Appendix A.2), and found that attention-based pooling yielded the best results among the pooling operators we considered. Moreover, to validate that the attention pooling learns to prioritize cells that most strongly reflect the perturbation effect, we performed a top-$n$ removal experiment: for the Bray et al. [4]. retrieval tasks, we removed the n samples with the highest attention weights for each perturbation and then recomputed retrieval performance. This led to a substantial performance drop (Table S.3), demonstrating that attention pooling effectively identifies and emphasizes the subset of cells carrying the strongest perturbation signal.

Overall, our results demonstrate that each of our main design choices in CellCLIP contributes to improved retrieval performance and computational efficiency compared to prior work.

## 5.2 Intra-modality retrieval

For our intra-modal retrieval benchmarking tasks, only image profile embeddings (and not perturbation embeddings) are required. Thus, to further contextualize our model's performance, for these tasks we also compared against OpenPhenom [27], a state-of-the-art unimodal self-supervised representation learning model for Cell Painting images. Specifically, for our experiments we used OpenPhenom-S/16, which was the only model openly released by Kraus et al. [27]. As in Kraus et al. [27], we also provide results for a unimodal weakly supervised vision transformer baseline trained to predict perturbation labels for further comparison. Finally, for these experiments we compared

against the best performing variants of CLOOME and MolPhenix* in our cross-modal retrieval results.

**Replicate detection and sister perturbation matching.**  We report results for these tasks in Table 4. We found that cross-modal CL outperformed our unimodal baselines. Moreover, within the considered CL approaches, we found that CellCLIP strongly outperformed all other approaches for replicate detection. For perturbation matching, when comparing perturbations targeting the same genes within the same perturbation class (e.g., compounds vs compounds), CellCLIP again performs the best among all approaches. For cross-class perturbation matching, (e.g. CRISPR vs compounds), CellCLIP again performs the best, though the overall performance for all machine-learning-based methods remains low. This suggests that, despite targeting the same gene, morphological changes across different perturbation classes remain highly distinct, aligning with previous findings [10].

**Recovering known biological relationships.**  Table 4 reports recall of known biological relationships among genetic perturbations in RxRx3-core. Across all benchmark databases, we found that CellCLIP achieved the best recovery of known gene-gene relationships compared to baseline models. Notably, we found that replacing the CWCL loss used in CellCLIP with the standard CLIP loss leads to worse performance on this task (Table S.5), illustrating the benefits of using soft labeling for alignment in the image profile space $\mathcal{P}$. Altogether, these results further demonstrate that CellCLIP can recover meaningful relationships between perturbations in its image profile latent space.

## 6   Conclusions

Here we addressed the challenge of representation learning for Cell Painting perturbation screens by introducing CellCLIP, a cross-modal CL framework that unifies perturbations across classes through textual descriptions. As part of our framework, we also developed CrossChannelFormer, a multi-instance, transformer-based architecture that efficiently captures channel dependencies and processes profile data while reducing computational costs. Our results demonstrate that CellCLIP improves cross-modal retrieval performance, intra-modal retrieval for downstream tasks, and generalization across perturbation types. Overall, CellCLIP offers a promising solution for analyzing high-content morphological screening data, and future work will explore the impact of various contrastive losses and the contributions of each Cell Painting channel to downstream performance.

**Limitation & Future Work**   While CellCLIP demonstrates strong performance, current evaluations are limited to datasets collected in previous works. Incorporating further validation of CellCLIP's results via additional wet lab experiments would better assess its real-world applicability. Another limitation is data accessibility: the high cost of generating Cell Painting images has resulted in many datasets and models remaining proprietary, limiting the reproducibility and benchmarking of models in this space more generally. Future work should prioritize the development of open, representative benchmarks. Finally, by aligning Cell Painting images with textual descriptions of perturbations, CellCLIP also opens new directions for conditional generation of phenotypic representations from natural language, paving the way for text-driven synthesis and simulation in drug discovery.

## 7   Acknowledgement

We thank members of the Lee lab for providing feedback on this project and the reviewers for their constructive comments. This work was funded by the National Science Foundation [DBI-1552309 and DBI-1759487]; and the National Institutes of Health [R01 AG061132, R01 EB035934, and RF1 AG088824].

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

Table S.1: Retrieval performance of CellCLIP trained with profiles generated from various pretrained imaging models on perturb-to-profile and profile-to-perturb tasks. Results are reported as Recall@$k$ (%) for $k = 1, 5, 10$.

| Image Encoding Backbone | # of Params | # of GFLOPs | Perturb-to-profile (%) | | | Profile-to-perturb (%) | | |
|---|---|---|---|---|---|---|---|---|
| | | | R@1 | R@5 | R@10 | R@1 | R@5 | R@10 |
| DINOv1 | 86.4M | 66.87 | $0.75 \pm 0.08$ | $2.77 \pm 0.13$ | $4.74 \pm 0.33$ | $0.86 \pm 0.16$ | $2.77 \pm 0.27$ | $4.80 \pm 0.19$ |
| DINOv2 (small) | 22.06M | 5.53 | $1.19 \pm 0.32$ | $4.06 \pm 0.27$ | $6.57 \pm 0.33$ | $\textbf{1.32} \pm \textbf{0.17}$ | $4.34 \pm 0.39$ | $6.63 \pm 0.19$ |
| DINOv2 (base) | 86.58M | 21.97 | $\textbf{1.19} \pm \textbf{0.12}$ | $4.03 \pm 0.20$ | $6.24 \pm 0.40$ | $1.25 \pm 0.12$ | $3.95 \pm 0.18$ | $6.32 \pm 0.38$ |
| DINOv2 (large) | 304.37M | 77.83 | $0.97 \pm 0.12$ | $4.33 \pm 0.43$ | $6.76 \pm 0.54$ | $1.16 \pm 0.33$ | $4.23 \pm 0.40$ | $6.50 \pm 0.47$ |
| **DINOv2 (giant)** | 1136.48M | 291.43 | $1.18 \pm 0.20$ | $\textbf{4.49} \pm \textbf{0.06}$ | $\textbf{7.37} \pm \textbf{0.20}$ | $1.25 \pm 0.10$ | $\textbf{4.82} \pm \textbf{0.10}$ | $\textbf{7.39} \pm \textbf{0.23}$ |
| OpenPhenom-S/16 | 178.05M | 83.42 | $0.94 \pm 0.04$ | $3.73 \pm 0.13$ | $6.22 \pm 0.20$ | $1.27 \pm 0.14$ | $4.13 \pm 0.19$ | $6.25 \pm 0.08$ |

Table S.2: Retrieval performance of CellCLIP trained across different pooling strategies on perturb-to-profile and profile-to-perturb tasks. Results are reported as Recall@$k$ (%) for $k = 1, 5, 10$.

| Pooling Strategy | Perturb-to-profile (%) | | | Profile-to-perturb (%) | | |
|---|---|---|---|---|---|---|
| | R@1 | R@5 | R@10 | R@1 | R@5 | R@10 |
| Attention-based | $\textbf{1.18} \pm \textbf{0.20}$ | $\textbf{4.49} \pm \textbf{0.06}$ | $\textbf{7.37} \pm \textbf{0.20}$ | $1.25 \pm 0.10$ | $\textbf{4.82} \pm \textbf{0.10}$ | $\textbf{7.39} \pm \textbf{0.23}$ |
| Median | $1.08 \pm 0.20$ | $4.03 \pm 0.39$ | $6.26 \pm 0.30$ | $1.04 \pm 0.13$ | $3.85 \pm 0.30$ | $6.00 \pm 0.13$ |
| Mean | $\textbf{1.18} \pm \textbf{0.12}$ | $4.45 \pm 0.23$ | $6.83 \pm 0.20$ | $\textbf{1.35} \pm \textbf{0.41}$ | $4.46 \pm 0.21$ | $6.69 \pm 0.28$ |
| K-means | $1.15 \pm 0.12$ | $4.03 \pm 0.19$ | $6.17 \pm 0.15$ | $1.04 \pm 0.07$ | $3.89 \pm 0.26$ | $6.06 \pm 0.12$ |
| Hierarchical clustering | $1.10 \pm 0.12$ | $4.08 \pm 0.09$ | $6.69 \pm 0.26$ | $1.12 \pm 0.29$ | $4.17 \pm 0.35$ | $6.55 \pm 0.15$ |

# Appendix

# A    Additional Results

## A.1    Effects of Different Image Profile Encoding

CellCLIP provides a flexible framework for integrating off-the-shelf pretrained image foundation models into Cell Painting analyses (Section 3.2). To understand the impact of different vision encoding backbones on CellCLIP's performance, we conducted an ablation study where we varied CellCLIP's image encoder while holding all other aspects of our framework constant. Specifically, for this experiment we considered DINOv1 along with DINOv2 models of varying sizes. Aligning with results for natural images, we found that increases in model size broadly led to increased performance on our retrieval tasks (Table S.1).

To understand the impact of using image encoder backbones originally trained on natural images versus those trained directly on Cell Painting data, we also applied OpenPhenom-S/16, an openly available masked autoencoder model pretrained on Cell Painting data [27]. Interestingly, we found that using OpenPhenom-S/16 did not result in superior performance compared to DINO models trained on natural images. This suggests that, despite not being originally trained on microscopy data, foundation models trained on diverse natural image distributions combined with small tweaks to account for differences in channels as in CellCLIP can achieve competitive performance on Cell Painting data.

## A.2    Effect of Various Pooling Strategy

Table S.2 summarizes the retrieval performance across different pooling strategies as described in Section 3.1. Overall, attention-based pooling yields the best performance, particularly on the Recall@10. Among non-attention methods, mean pooling consistently outperforms median, K-means, and hierarchical clustering, suggesting its effectiveness as a simple yet strong baseline.

## A.3    Impact of top-$n$ instances

To mitigate the influence of false positives, we applied per-perturbation pooling of cellular profiles using the ABMIL pooling operator [24], which adaptively attends to samples most representative of a perturbation. We further investigated the importance of accounting for false positives by removing the most highly attended samples (i.e., those most indicative of effective perturbation). cross-modal retrieval tasks. This ablation led to substantial performance degradation (see Table S.3), suggesting that for each perturbation, only a relatively small fraction of cells exhibit strong perturbation effects.

Table S.3: Retrieval performance after removing the top-$n$ contributing instances. Results are reported as Recall@$k$ (%) with $k = 1, 5, 10$.

| Removal of top-$n$ instances | Perturb-to-profile (%) | | | Profile-to-perturb (%) | | |
|---|---|---|---|---|---|---|
| | R@1 | R@5 | R@10 | R@1 | R@5 | R@10 |
| 0 | 1.32 | 4.59 | 7.51 | 1.18 | 4.82 | 7.56 |
| 1 | 1.08 | 4.60 | 7.13 | 1.04 | 4.58 | 7.61 |
| 2 | 1.13 | 4.25 | 6.38 | 0.99 | 4.34 | 6.47 |
| 3 | 1.13 | 4.20 | 5.91 | 0.89 | 4.06 | 6.09 |
| 4 | 0.04 | 2.50 | 5.01 | 0.09 | 3.76 | 5.34 |
| 5 | 0.00 | 1.46 | 3.68 | 0.00 | 2.88 | 4.34 |

Table S.4: Retrieval performance of CellCLIP trained across different loss functions on perturb-to-profile and profile-to-perturb tasks. Results are reported as Recall@$k$ (%) for $k = 1, 5, 10$.

| Loss type | Perturb-to-profile (%) | | | Profile-to-perturb (%) | | |
|---|---|---|---|---|---|---|
| | R@1 | R@5 | R@10 | R@1 | R@5 | R@10 |
| CLIP [35] | $\mathbf{1.26 \pm 0.10}$ | $4.24 \pm 0.06$ | $7.01 \pm 0.41$ | $\mathbf{1.29 \pm 0.19}$ | $4.15 \pm 0.31$ | $6.55 \pm 0.29$ |
| CLOOB [17] | $0.87 \pm 0.07$ | $3.32 \pm 0.36$ | $5.39 \pm 0.45$ | $0.91 \pm 0.14$ | $3.52 \pm 0.26$ | $5.51 \pm 0.43$ |
| SigCLIP [48] | $1.10 \pm 0.08$ | $3.54 \pm 0.20$ | $6.03 \pm 0.30$ | $1.07 \pm 0.11$ | $3.79 \pm 0.09$ | $6.22 \pm 0.11$ |
| S2L [16] | $1.04 \pm 0.19$ | $3.93 \pm 0.45$ | $6.83 \pm 0.50$ | $1.08 \pm 0.10$ | $4.25 \pm 0.14$ | $6.65 \pm 0.21$ |
| CWCL [41] | $1.18 \pm 0.20$ | $\mathbf{4.49 \pm 0.06}$ | $\mathbf{7.37 \pm 0.20}$ | $1.25 \pm 0.10$ | $\mathbf{4.82 \pm 0.10}$ | $\mathbf{7.39 \pm 0.23}$ |

Table S.5: Comparison of CellCLIP with different contrastive losses for zero-shot biological relationship recovery (RxRx3-core). Results report average recall across cosine similarity thresholds (5th and 95th percentiles).

| Loss type | Gene-Gene Relationship Recovery (Recall) ↑ | | | | |
|---|---|---|---|---|---|
| | CORUM | HuMAP | Reactome | SIGNOR | StringDB |
| CLIP | .673 | .751 | .416 | .370 | .595 |
| CLOOB | .696 | .772 | .418 | .373 | .598 |
| SigCLIP | **.741** | **.800** | **.466** | **.454** | **.646** |
| S2L | .691 | .765 | .416 | .385 | .603 |
| CWCL | .714 | .778 | .427 | .388 | .618 |

## A.4 Effect of Different Contrastive Loss

Table S.4 summarizes cross-modal retrieval performance across different contrastive learning losses. CWCL consistently achieves the highest Recall@5 and Recall@10 in both perturbation-to-profile and profile-to-perturbation tasks, while remaining competitive at Recall@1. CLIP ranks second overall, and CLOOB shows the weakest performance. Although both CWCL and S2L use similarity-based weighting, CWCL outperforms S2L across all retrieval metrics. This suggests that computing similarity from embeddings extracted from pretrained imaging foundation models is more effective for alignment than using projected embeddings in the shared space, as defined in Fradkin et al. [16].

Table S.5 presents intra-modal gene–gene recovery performance. Sigmoid loss (SigCLIP) achieves the highest recall across all benchmarks, followed by CWCL. This indicates that contrastive objectives with relaxed pairwise penalties may better preserve biological structure. Overall, CWCL demonstrates strong performance across both settings, highlighting its ability to balance cross-modal alignment with intra-modal consistency.

Table S.6: Comparison of models for zero-shot biological relationship recovery (RxRx3-core). Results report average recall across cosine similarity thresholds (5th and 95th percentiles).

| Method | Gene-Gene Relationship Recovery (Recall) ↑ | | | | |
| --- | --- | --- | --- | --- | --- |
| | CORUM | HuMAP | Reactome | SIGNOR | StringDB |
| *Cross-modal contrastive learning* | | | | | |
| CLOOME | .597 | .679 | .327 | .309 | .510 |
| Molphenix* | .539 | .599 | .330 | .297 | .476 |
| CellCLIP (DINOv1) | .720 | .774 | .445 | .378 | .614 |
| CellCLIP (DINOv2-Small) | .709 | .777 | .427 | .400 | .618 |
| CellCLIP (DINOv2-Base) | .706 | .766 | .446 | .406 | .616 |
| CellCLIP (DINOv2-Large) | **.725** | **.780** | **.450** | **.406** | **.632** |
| CellCLIP (DINOv2-Giant) | .714 | .778 | .427 | .388 | .618 |
| *HCS-pretrained Channel-agnostic MAE* | | | | | |
| OpenPhenom-S/16 | .649 | .723 | .418 | .386 | .579 |
| *ImageNet-pretrained classifiers / weakly supervised models* | | | | | |
| ViT-L/16 | .681 | .758 | .388 | .380 | .587 |

## A.5 Zero-shot Biological Relationship Recovery - RxRx3-core

We evaluate recall over the most extreme 2% to 20% of the similarity distribution on RxRx3-core (Figure S.1), including pathway annotations from CORUM, HuMAP, Reactome, SIGNOR, and STRING databases. As shown in the figure below, CellCLIP consistently achieves the highest recall across thresholds, outperforming existing multimodal contrastive methods such as CLOOME and MolPhenix. Among CellCLIP variants, performance is comparable across different image tokenizers, indicating robustness to the choice of visual backbone (Table S.6).

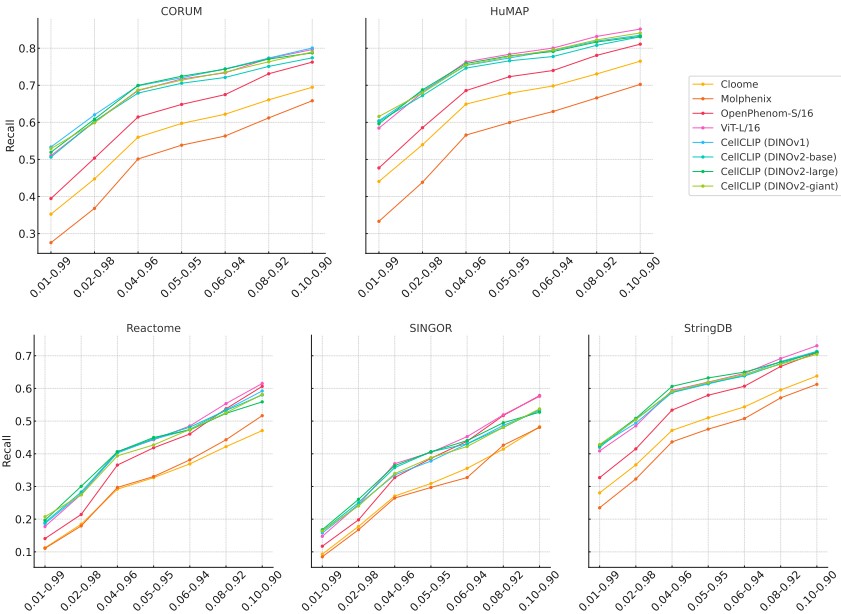

Figure S.1: Zero-shot gene–gene relationship recovery on RxRx3-core, evaluated across varying thresholds from Recall@2% [0.01,0.99] (top and bottom 1%) to Recall@20% [0.10,0.90] (top and bottom 10%), using pathway annotations from CORUM, HuMAP, Reactome, SIGNOR, and STRING.

## A.6 Effect of Text Encoder Parameters

In Table S.7, we present additional experiments using smaller BERT variants (64M and 11.3M parameters) from Turc et al. [44] as the text encoder for CellCLIP on the cross-modal retrieval task [4]. For reference, the MPNN+ graph neural network encoder in our MolPhenix baseline contains approximately 10M parameters. While performance decreases slightly with smaller BERT models, even the 11.3M-parameter variant substantially outperforms the MPNN+ baseline. These results

Table S.7: Retrieval performance with different BERT parameter sizes. Results are reported as Recall@$k$ (%) with $k = 1, 5, 10$.

| # BERT params | Perturb-to-profile (%) | | | Profile-to-perturb (%) | | |
|---|---|---|---|---|---|---|
| | R@1 | R@5 | R@10 | R@1 | R@5 | R@10 |
| 110M (ours) | $1.18 \pm 0.20$ | $4.49 \pm 0.06$ | $7.37 \pm 0.20$ | $1.25 \pm 0.10$ | $4.82 \pm 0.10$ | $7.39 \pm 0.23$ |
| 64M | $1.53 \pm 0.17$ | $4.40 \pm 0.21$ | $6.90 \pm 0.18$ | $1.41 \pm 0.17$ | $4.67 \pm 0.24$ | $6.98 \pm 0.14$ |
| 11.3M | $1.13 \pm 0.14$ | $4.04 \pm 0.31$ | $6.28 \pm 0.19$ | $1.28 \pm 0.17$ | $4.11 \pm 0.24$ | $6.20 \pm 0.19$ |

Table S.8: Retrieval performance with different perturbation encoders. Results are reported as Recall@$k$ (%) with $k = 1, 5, 10$.

| Perturb. Encoder | Perturb-to-profile (%) | | | Profile-to-perturb (%) | | |
|---|---|---|---|---|---|---|
| | R@1 | R@5 | R@10 | R@1 | R@5 | R@10 |
| BiomedBERT | $1.46 \pm 0.10$ | $4.11 \pm 0.13$ | $6.19 \pm 0.26$ | $1.51 \pm 0.09$ | $4.34 \pm 0.13$ | $6.57 \pm 0.19$ |
| KV-PLM | $1.29 \pm 0.19$ | $4.41 \pm 0.31$ | $6.92 \pm 0.32$ | $1.41 \pm 0.16$ | $4.57 \pm 0.30$ | $6.81 \pm 0.21$ |
| BERT | $1.18 \pm 0.20$ | $4.49 \pm 0.06$ | $7.37 \pm 0.20$ | $1.25 \pm 0.10$ | $4.82 \pm 0.10$ | $7.39 \pm 0.23$ |

indicate that the gains in cross-modal retrieval are not merely due to larger model capacity, but rather arise from the effectiveness of our text-based encoding strategy.

## A.7    Do Domain-Adapted BERT Help?

In Table S.8, we compare several pretrained BERT models, including BiomedBERT [49], trained on biomedical PubMed corpora, and KV-PLM [47], a BERT-based model pretrained on both biomedical texts and SMILES strings. Interestingly, while KV-PLM outperforms BiomedBERT, it still underperforms relative to our original BERT choice, suggesting that domain adaptation alone does not guarantee superior performance for cross-modal retrieval.

Table S.9: Image foundation models used to generate per-channel embeddings, along with their parameter sizes and total embedding extraction time for Bray et al. [4]. Extraction times were measured using a single NVIDIA A40 GPU.

| Model | Parameter Count (M) | Extraction Time (hr) |
|---|---|---|
| DINO (ViT-B/16) | 86.4 | 5.21 |
| DINOv2 (ViT-S/14) | 22.06 | 1.83 |
| DINOv2 (ViT-B/14) | 86.58 | 2.78 |
| DINOv2 (ViT-L/14) | 304.37 | 5.22 |
| DINOv2 (ViT-G/14) | 1136.48 | 16.3 |
| CA-MAE (OpenPhenom-S/16) | 178.05 | 2.89 |

# B  CellCLIP Training & Implementation Details

**Feature extractors for Cell Painting profiles**    To generate channel profiles $z_k$ (Section 3.1), we experimented with a range of image foundation models, including DINO, DINOv2 (small, base, large, giant), and CA-MAE (OpenPhenom-S/16) (Table S.9). All model checkpoints were obtained from Hugging Face. Cell Painting images were treated as grayscale inputs, preprocessed according to the requirements of each model, and augmented using a multi-crop strategy to enhance robustness.

**Perturbation encoding**    For generating text prompts, we adopted the following template:

> *"A {cell type} treated with {perturbation},*
> *with {detailed perturbation information, such as SMILES or target genes}"*.

We also experimented with alternative templates, such as:

> *"A cell painting image of {cell type} treated with {perturbation},*
> *with {detailed perturbation information, such as SMILES or target genes}"*,

as well as other variants, but observed no significant difference in cross-modal retrieval performance.

**Attention-based multi-instance learning (ABMIL).**    To accommodate the varying number of Cell Painting images per perturbation, we introduce a multi-instance learning (MIL) pooling mechanism [24] in CrossChannelFormer. Instead of relying on a single instance, CrossChannelFormer processes multiple input profiles associated with the same perturbation and aggregates them into a single representation.

Formally, given a perturbation $i$ and its associated set of image profiles $\{z_k\}_{k=1}^{N_i}$, we define the pooled profile $\mu(U_i)$ as follows:

$$\mu(U_i) = \left\{ \mu(U_i)^1, \mu(U_i)^2, \ldots, \mu(U_i)^C \right\} \in \mathbb{R}^{C \times d}, \tag{7}$$

where $\mu(U_i)^c$ denotes the aggregated representation for channel $c \in \{1, \ldots, C\}$.

Each channel-wise pooled representation is obtained via a permutation-invariant aggregation function $\mathcal{S}(\cdot)$:

$$\mu(U_i)^c = \mathcal{S}\left(\{z_k^c \mid k = 1, \ldots, N_i\}\right) \in \mathbb{R}^d, \tag{8}$$

The aggregation function $\mathcal{S}(\cdot)$ can be instantiated as any permutation-invariant transformation function applied channel-wise, such as maximum or mean operator. To better capture variability among instances, we adopt gated attention pooling [24], defined as

$$\mu(U_i)^c = \sum_{k=1}^{N_i} \alpha_k^c z_k^c \in \mathbb{R}^d, \quad \text{where}$$

$$\alpha_k^c = \frac{\exp\left(\mathbf{w}^\top \left(\tanh(\mathbf{V} z_k^c) \odot \sigma(\mathbf{U} z_k^c)\right)\right)}{\sum_{k'=1}^{N_i} \exp\left(\mathbf{w}^\top \left(\tanh(\mathbf{V} z_{k'}^c) \odot \sigma(\mathbf{U} z_{k'}^c)\right)\right)} \tag{9}$$

Here, $\mathbf{w} \in \mathbb{R}^L$ and $\mathbf{U}, \mathbf{V} \in \mathbb{R}^{L \times d}$ are learnable parameters, $\odot$ denotes element-wise multiplication, and $\sigma(\cdot)$ is the sigmoid activation function. While we adopt attention-based pooling in this work to model instance-level variability, the aggregation function $\mathcal{S}(\cdot)$ is flexible and can be instantiated as simpler alternatives, such as mean pooling [16].

**CellCLIP backbone**    Our CrossChannelFormer backbone consists of a transformer model with 12 layers, 8 attention heads, and a 512-dimensional embedding space. For the text encoder, we employ the pre-trained BERT[12]. It supports text lengths of up to 512 and uses WordPiece [29] as its tokenizer.

**Training details**    For retrieval evaluation on Bray et al. [4], CellCLIP was trained for 50 epochs with a batch size of 768 using the AdamW optimizer. The learning rate was set to $2 \times 10^{-4}$ with cosine annealing and restarts. The temperature parameter $\tau$ was initialized as 14.3.

For RxRx3-core, we performed zero-shot evaluation by reusing the models trained on Bray et al. [4] to assess their ability to recover known biological relationships.

For CP-JUMP1, we reuse the model trained with Bray et al. [4] and further fine-tune with CP-JUMP1. Fine-tuning was performed for an additional 50 epochs using the same hyperparameter settings as in pretraining.

# C Experiment & Evaluation Metrics Details

## C.1 Cross-Modal Retrieval

We evaluate cross-modality retrieval using Recall@k, a standard metric that measures the proportion of queries whose correct match appears in the top $k$ retrieved items. Given a batch of $N$ paired samples $\{(u_i, v_i)\}_{i=1}^N$, where $u_i \in \mathcal{U}$ and $v_i \in \mathcal{V}$ are Cell Painting profiles and perturbation and trained encoder $f_\theta : \mathcal{U} \to \mathcal{P}$ and $g_\phi : \mathcal{V} \to \mathcal{Q}$, we compute the similarity matrix $S \in \mathbb{R}^{N \times N}$, where

$$S_{ij} = \frac{\langle p_i \cdot q_j \rangle}{\|p_i\| \, \|q_j\|}. \tag{10}$$

Recall@K in the $\mathcal{U} \to \mathcal{V}$ direction (e.g., profile-to-molecules) is defined as:

$$\text{Recall@k}_{\mathcal{U} \to \mathcal{V}} = \frac{1}{N} \sum_{i=1}^N \mathbb{1} \left[ \text{rank}_{\mathcal{V}}(q_i \mid p_i) \leq k \right], \tag{11}$$

where $\text{rank}_{\mathcal{V}}(p_i \mid q_i)$ is the rank of the correct match $q_i$ when all $q_j$ are sorted by similarity to $p_i$, and $\mathbb{1}[\cdot]$ is the indicator function. The reverse direction $\mathcal{V} \to \mathcal{U}$ is defined analogously. In this work, we report Recall@k with $k \in \{1, 5, 10\}$.

## C.2 Biological Evaluation of Learned Representations via Intra-Modal Retrieval

Here we provide more details for *intra-modality* retrieval for Cell Painting analysis [9, 10, 27]. These tasks include (1) replicate detection, (2) sister perturbation matching and (3) zero-shot recovery of known biological relationships.

### C.2.1 Replicate Detection (CP-JUMP1 [10])

For replicate detection, we follow the protocol of Chandrasekaran et al. [10] and Kalinin et al. [25]. In particular, during model training and evaluation we pool images across both perturbations $i$ and experimental batches $s$ (corresponding to e.g. different well positions). That is, for each combination of perturbation $i$ and batch $s$, we obtain an embedding $p_i^s$. Each $p_i^s$ is then corrected for batch effects using matched negative controls from the same experimental batch $s$ to produce a corrected embedding $\tilde{p}_i^s$.

Given these batch-effect-corrected pooled embeddings, we compute cosine similarities between a query $\tilde{p}_i^s$ and candidate embeddings $\tilde{p}_j^{s'}$ corresponding to replicates of the same perturbation $i$ or negative controls, and then rank candidates in descending order. Following Chandrasekaran et al. [10], we report *average precision* (AP) as our evaluation metric for this task:

$$AP = \sum_{k=1}^n (R_k - R_{k-1}) P_k \tag{12}$$

where $P_k$ and $R_k$ denote the precision and recall at rank $k$, respectively. To assess statistical significance, we perform permutation testing by shuffling rankings 100,000 times to construct a null distribution. We then apply multiple comparison corrections [3] and filter out non-significant AP values. For each perturbation replicate in different batches, we compute AP scores, which are then averaged to obtain a *mean average precision (mAP)* score representing the perturbation's phenotypic activity. Finally, we use mAP across classes, defined by specific perturbations or gene associations, to evaluate the performance in both tasks.

Our evaluation pipeline wes implemented following the official benchmark protocol[4].

### C.2.2 Sister Perturbation Matching (CP-JUMP1 [10])

For sister perturbation matching, we again follow the protocols of Chandrasekaran et al. [10] and Kalinin et al. [25]. However, after computing batch-effect-corrected pooled embeddings $\tilde{p}_i^s$ as

---

[4]Benchmark pipeline for CP-JUMP1

described above, we now perform a second aggregation step across experimental conditions to produce a single $\tilde{p}_i$ for each perturbation across batches. That is, we compute

$$\tilde{p}_i = \mathcal{A}(\{\tilde{p}_i^s\}_{s=1}^{n_s}),\tag{13}$$

where $\mathcal{A}$ is some permutation-invariant aggregation function. For these experiments we used the mean operation to aggregate across batches.

Using these aggregated results, we then similarly we compute cosine similarities between a query $\tilde{p}_i$ and candidate embeddings $\tilde{p}_j$, and rank candidates in descending order. As described above we again use mean average precision across classes as our evaluation metric for this task.

### C.2.3 Zero-shot Recovery of Known Biological Relationship (RxRx3-core [28])

This task evaluates whether the learned embeddings capture biologically meaningful structure by assessing their ability to recover known functional relationships among perturbations, such as genes within the same pathway or protein complex. Pathway annotations are obtained from HuMAP, CORUM, StringDB, Reactome, and SIGNOR (Table S.10).

For each perturbation $i$, we compute an aggregated embedding $\tilde{p}_i$ as described in Appendix C.2.2 but using the median for $\mathcal{A}$ as in Kraus et al. [27]. For each perturbation $i$ we then evaluate the cosine similarity of $\tilde{p}_i$ with embeddings $\tilde{p}_j$ from other perturbations. All perturbation pairs are then ranked by similarity to assess the recovery of known biological relationships.

Recall is then evaluated over the most extreme 10% of the similarity distribution, including the top 5% (most similar pairs) and bottom 5% (most dissimilar pairs), and is defined as:

$$\text{Recall@10\%} = \frac{\text{\# of known positives recovered in top \& bottom 5\%}}{\text{Total \# of known positives}}.\tag{14}$$

A random embedding space yields a baseline recall of 10%. Higher recall indicates stronger alignment with curated biological interaction networks. We also report recall@2% ([0.01, 0.99]) to recall@ 20% ([0.10, 0.90]) in Appendix A.5. Evaluation was performed following the official RxRx3-core benchmark protocol.[5]

### C.3 Batch Effect Correction

For the batch correction operations mentioned above, we followed the procedure of Celik et al. [9] implemented in 5 for RxRx3-core and CP-JUMP1. A PCA kernel[6] was fit on all control profiles across experimental batches and used to transform all embeddings. For each batch, a separate `StandardScaler` was fit on the transformed control embeddings and applied to normalize all embeddings. We experimented with RBF, polynomial, and linear kernels and selected the best-performing kernel for each method.

---

[5]EFAAR Benchmark

[6]Scikit-learn KernelPCA

# D    Datasets & Preprocessing

**Bray et al. [4]**    The dataset[7] consists of 919,265 five-channel microscopy images with resolution $520 \times 696$ corresponding to 30,616 different molecules. These images were captured using 406 multi-well plates, with each image representing a view from a sample within one well. Six adjacent views collectively form one sample.

Sanchez-Fernandez et al. [38] refined the dataset by removing images that were out of focus, exhibited high fluorescence, or contained untreated control cells. The final dataset comprises 28.4k images linked to 10.8k unique molecules, split into training, validation, and test sets. The processed dataset is publicly available at[8]. Next, for retrieval evaluation on unseen compounds (Section 4.1), following Sanchez-Fernandez et al. [38], we removed samples from the test set that corresponded to the same molecule and plate to mitigate plate effects. The remaining samples are referred to as the final "test set", which consists of 2,115 unique compounds.

**CP-JUMP1**    CP-JUMP1 [10] comprises approximately 340k eight-channel (3 bright field channels) microscopy images of resolution $1080 \times 1080$. We use images from the batch named `2020_11_04_CPJUMP1`, which contains 186,925 images. This dataset features a comprehensive collection of perturbations conducted on U2OS and A549 cell lines, including 52 replicates. For perturbations, it includes 301 small-molecule compounds (46 controls), 335 sgRNAs (CRISPR) targeting 175 genes (88 control sgRNAs), and 175 ORFs (45 controls) for the corresponding genes. [10] also provides annotations of associations between genes and compounds, enhancing its utility for exploring gene function and compound effects. We split the dataset into 70/10/20 for training, validation, and testing. Raw data, relevant metadata, and gene annotation can be found in[9].

**RxRx3-core**    RxRx3-core[28][10] is a curated subset of the RxRx3 dataset [15]. It includes 222,601 six-channel fluorescent microscopy images of HUVEC cells stained using a modified Cell Painting protocol, covering 736 gene knockouts (non-blinded) and 1,674 compounds across eight concentrations, along with control wells. Gene-gene relationships are benchmarked using Reactome, HuMAP, SIGNOR, StringDB, and the CORUM collection, Table S.10. Additional details regarding RxRx3-core can be found in Kraus et al. [28].

Table S.10: Description of datasets used for curating gene-gene interactions

| Datasets | Description |
| --- | --- |
| Reactome [18] | Protein-protein interactions based on curated pathways. |
| StringDB [43] | Proteins associations based on functional interactions. |
| HuMAP [14] | Gene clusters representing protein complexes. |
| CORUM [19] | Gene clusters corresponding to protein complexes. |
| SIGNOR [34] | Protein-protein interactions in signaling pathways. |

**Image Preprocessing**    For both datasets, our preprocessing followed the protocols established by Sanchez-Fernandez et al. [38] and Hofmarcher et al. [22], which consist of converting the original TIF images from 16-bit to 8-bit and removing the 0.0028 % of pixels with the highest values[11]. To maintain channel consistency between datasets, we processed the CP-JUMP1 images and reordered channels to match the five-channel format of [4].

---

[7]GigaDB doi.org/10.5524/100351

[8]Curated dataset of Bray et al. [4]

[9]Github repository of Chandrasekaran et al. [10]

[10]RxRx3-core dataset

[11]TIF files preprocessing

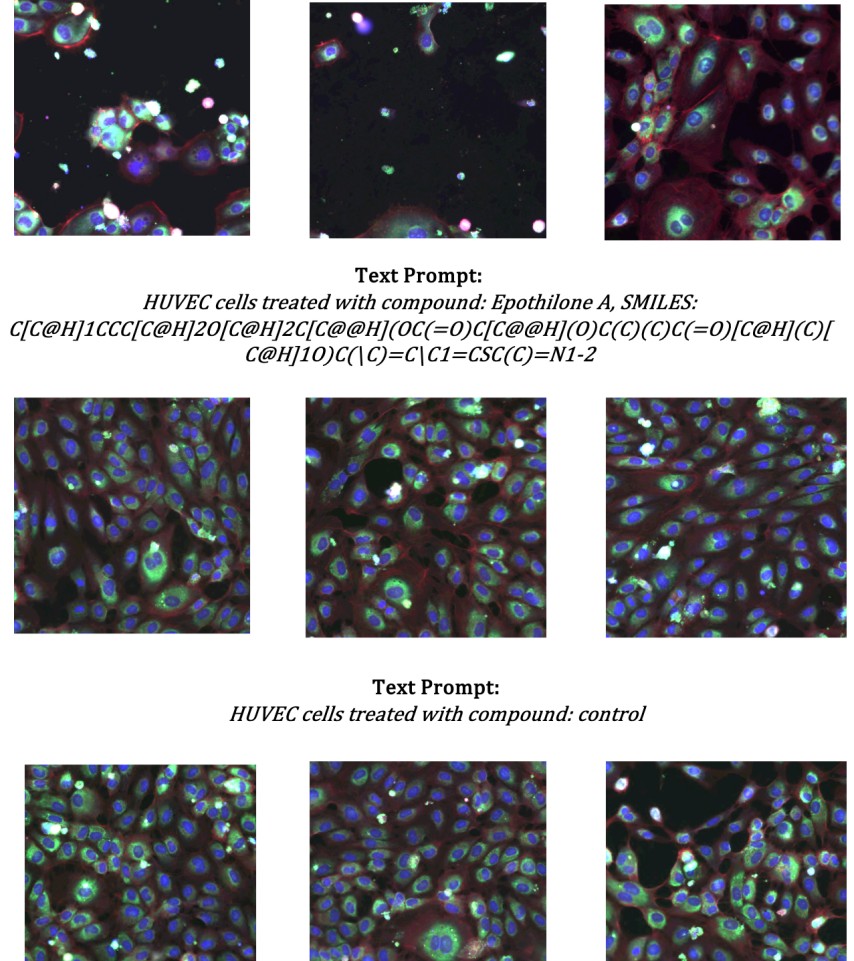

**Text Prompt:**
*HUVEC cells treated with compound: Epothilone A, SMILES:*
*C[C@H]1CCC[C@H]2O[C@H]2C[C@@H](OC(=O)C[C@@H](O)C(C)(C)C(=O)[C@H](C)[*
*C@H]1O)C(\C)=C\C1=CSC(C)=N1-2*

**Text Prompt:**
*HUVEC cells treated with compound: control*

**Text Prompt:**
*HUVEC cells treated with crispr, targeting genes RPL27A*

Figure S.2: Example Cell Painting images and corresponding prompts in RxRx3-core.

# E  Baselines Training & Implementation details

## E.1  Baselines

**CLOOME**  We follow the official CLOOME implementation[12]. For retrieval evaluation on the dataset from Bray et al. [4], we adopt the best-performing hyperparameters reported in Sanchez-Fernandez et al. [38], using a ResNet-50 as the vision encoder and a four-layer MLP as the molecule encoder, excluding the Hopfield layer. The model is trained using the original contrastive loss, with raw Cell Painting images paired with a max-pooled combination of Morgan and RDKit count-based fingerprints[13], resulting in an 8192-bit input representation. We refer to the variant that incorporates the Hopfield layer for CLOOB loss computation, as described in Sanchez-Fernandez et al. [38], as CLOOME (CLOOB).

The training setup includes a batch size of 256, the AdamW optimizer, and a learning rate of $1 \times 10^{-3}$ with cosine annealing and restarts. The learnable temperature parameter $\tau$ is initialized to 14.3. The model is trained for 70 epochs. We use the same training configuration for all CLOOME variants. For CP-JUMP1, since CLOOME is originally designed for small molecules, we fine-tune the model using only the small-molecule subset of the CP-JUMP1 training set, with the same hyperparameters as used for training on Bray et al. [4].

**CLOOME$^{\ddagger}$**  Following Fradkin et al. [16], we refer to the variant of CLOOME that uses phenomic profiles—specifically, the mean-pooled embeddings for each perturbation—as input, as CLOOME$^{\ddagger}$. For the profile modality, this variant replaces the vision encoder with fixed phenotypic embeddings while retaining CLOOME's original molecule encoder for chemical compounds.

**MolPhenix***  As the official codebase for MolPhenix is not publicly available, we re-implement the model based on the descriptions provided in Fradkin et al. [16]. The original feature extractors for phenomic (profile) and molecular inputs—Phenom1 [27] and MolGPS [42]—are either trained on proprietary datasets or lack open-source implementations. Therefore, we substitute these components with publicly available alternatives.

For phenomic features, we use OpenPhenom-S/16 as the feature extractor. For molecular features, we implement a 16-layer MPNN++ model trained on the PCQM4M_G25_N4 dataset [23] for 40 epochs, which has demonstrated comparable performance to MolGPS [42]. Using these pre-extracted features, we reconstruct MolPhenix according to the architecture reported in the original paper: six ResNet blocks followed by a linear layer for the phenomic modality, and one ResNet block followed by a linear layer for the molecular modality. The embedding dimension is set to 512. The embedding dimension is set to 512. The model is trained with S2L loss which applies an `arctan` distance function to projected embeddings in the shared latent space, using a similarity clipping threshold of 0.75, and hyperparameters $\gamma = 1.75$ and $\delta = 0.75$, consistent with the original work. We also experimented with the sigmoid loss introduced by Zhai et al. [48] and found it outperforms S2L in our setting. Accordingly, we adopt sigmoid loss as the default objective. Given possible differences from the original implementation, we refer to our version as MolPhenix*.

For CP-JUMP1, MolPhenix* was fine-tuned using only small molecules from the CP-JUMP1 training set, following the same training parameters as for Bray et al. [4].

**Weakly Supervised Learning (WSL)**  Following Kraus et al. [27], we constructed a Vision Transformer (ViT) Large with a patch size of 16 (ViT-L/16[14]), modified to accommodate five input channels [1]. A classifier head was attached, and the model was trained for 10 epochs with a learning rate of $1 \times 10^{-3}$ and weight decay. The batch size was set to 256. We used the output from the penultimate layer as the learned embeddings for evaluation.

**Channel-Agnostic Masked Autoencoder (CA-MAE) OpenPhenom-S/16**  CA-MAE [27] is a channel-agnostic image encoder based on a ViT-S/16 backbone, specifically designed for microscopy image featurization. It employs a vision transformer with channel-wise cross-attention over patch tokens to generate contextualized representations for each channel independently. For our evaluation,

---

[12]CLOOME's GitHub repository
[13]The official sources for the RDKit library
[14]PyTorch Image Models

Table S.11: Hyperparameter search range for cross-modal retrieval.

| Hyperparameter | Range / values |
| --- | --- |
| Learning rate | $5 \times 10^{-5}$, $1 \times 10^{-5}$, $1 \times 10^{-4}$, $2 \times 10^{-4}$, $3 \times 10^{-4}$, $5 \times 10^{-4}$, $1 \times 10^{-3}$, $1 \times 10^{-2}$ |
| Warmup steps | 200, 500, 800, 1000, 1500 |
| Cosine-annealing restart cycles | 1, 2, 3, 4 |
| Temperature ($\tau$) | 14.3, 20, 30 |

Table S.12: Hyperparameter search range for CellCLIP ablations.

| Hyperparameter | Range / values |
| --- | --- |
| Epochs | 50, 60, 70 |
| Learning rate | $5 \times 10^{-4}$, $10^{-3}$, $5 \times 10^{-3}$ |
| Cosine-annealing restart cycles | 5, 6, 7, 8, 9, 10 |
| Temperature ($\tau$) | 14.3, 20, 30 |

Table S.13: Hyperparameter search range for sister perturbation matching and replicate detection.

| Hyperparameter | Range / values |
| --- | --- |
| Epochs | 5, 10, 20, 30, 40, 50, 60, 70 |
| Batch size | 128, 256, 512 |
| Learning rate | $5 \times 10^{-5}$, $1 \times 10^{-5}$, $1 \times 10^{-4}$, $2 \times 10^{-4}$, $3 \times 10^{-4}$, $5 \times 10^{-4}$, $1 \times 10^{-3}$, $1 \times 10^{-2}$ |
| Warmup steps | 500, 800, 1000 |

we use the only publicly available version OpenPhenom-S/16. Since OpenPhenom-S/16 is already pretrained on CP-JUMP1 and RxRx3, we use the pretrained encoder to extract image embeddings without performing any additional fine-tuning.

### E.2 Hyperparameter Search

For CellCLIP, CLOOME (Phenomic), and MolPhenix* results in Table 1, we perform hyperparameter search over Table S.11. For batch size, we use the largest possible setting on 8 RTX 6000 GPUs: CellCLIP, MolPhenix*, and CLOOME (Phenomic) are trained with a batch size of 512, 768, and 768, while CLOOME is trained with a batch size of 256.

For the CellCLIP ablations reported in Table 2, we sweep the ranges in Table S.12. For the sister perturbation matching and replicate detection experiments (Table 4), we sweep the ranges in Table S.13.

For both of the unimodal CA-MAE and OpenPhenom-S/16 models specifically, as these models' training data already included CP-JUMP1, we did not perform any additional training steps with these models

### E.3 Software & Hardware Details

This study employs the PyTorch package tutorial (version 2.2.1). All experiments are conducted on systems equipped with 64 CPU cores and the specified NVIDIA GPUs. Models were trained with the largest possible batch size on 8 RTX 6000 GPUs.

