# OpenReview forum: "CellCLIP - Learning Perturbation Effects in Cell Painting via Text-Guided Contrastive Learning"
_NeurIPS.cc/2025/Conference — NeurIPS 2025 poster_

### Official Review · Reviewer_Hzy8 · 2025-06-06

**Clarity:** 3
**Significance:** 2
**Originality:** 4
**Rating:** 4
**Confidence:** 2

**Summary:**

This paper introduces CellCLIP, a cross-modal contrastive learning framework to capture the connections between Cell Painting image profiles and textual descriptions of cell perturbation effects. CellCLIP develops a CrossChannelFormer architecture to combine representations from multiple channels encompassing independent information. CellCLIP addresses the false negative problem by combining different Cell Painting image profiles with the same perturbation effect with a pooling layer, and introducing a weighted term that penalizes pushing similar profiles too far. Experiment results show that CellCLIP significantly outperforms existing methods in both cross-modal retrieval and intra-modal retrieval tasks.

**Questions:**

My major concerns are listed in Weaknesses. Here are a few minor questions:
- How does each component (cell type, perturbation, and detailed perturbation information) of the designed prompt contribute to the overall performance? I speculate that there could be some interesting observations and insights.
- Why do authors choose BERT for text encoding? Will domain-adapted language models trained on scientific literature and molecular SMILES, like BiomedBERT [1] and KV-PLM [2] bring further benefits?
- The CrossChannelFormer consists of 12 layers to encode C+1 features, where C is no greater than 10. I'm not sure if a deep model is required to encode such a short sequence.
- The model is pretrained on [3] for 50 epochs and fine-tuned on CPJUMP1 for another 50 epochs, raising concerns on catastrophic forgetting. Do the authors notice the benefits of transfer learning?

I promise to raise my scores to 4 if my concerns are addressed satisfactorily.

References

[1] BioMedBERT: A Pre-trained Biomedical Language Model for QA and IR.

[2] A deep-learning system bridging molecule structure and biomedical text with comprehension comparable to human professionals.

[3] Cell painting, a high-content image-based assay for morphological profiling using multiplexed fluorescent dyes.

**Ethical Concerns:**

["NO or VERY MINOR ethics concerns only"]

**Final Justification:**

The paper is well-motivated, featuring solid technical designs and thorough experimental evaluations. My primary concerns regarding the clarity and motivation of the work—particularly with respect to the issues of false positives and false negatives—were adequately addressed during the rebuttal phase. Overall, I consider this to be a high-quality submission and recommend it for publication.

That said, I am not deeply familiar with the task of perturbation effect prediction, and therefore I am uncertain about the broader impact of the paper on the wider research community. This limits my confidence in assigning a higher score. If fractional scores (e.g., 4.5) were permitted, I would have assigned such a score.

**Limitations:**

Yes.

**Quality:**

4

**Strengths And Weaknesses:**

Strengths
- The motivation of the paper, i.e., the multi-channel challenge, the false negative problem, and the false positive problem, is clear and closely related to real-world scientific applications.
- The introduced CrossChannelFormer architecture is intuitive and efficient.
- The experiment results show significant improvements over prior works. The analyses of the choice of encoding backbones, pooling strategies, and contrastive objectives are quite comprehensive.

Weaknesses
- I'm not fully convinced that the false negatives and false positives are real existing problems in HCS data. Quantitative and qualitative analyses on the data sources are required to justify the existence of the problem and that the proposed method mitigates the problem.
- The authors suggest using texts instead of molecular structures to encode cell perturbations to enhance the transferability of the model. While this makes sense, I'm wondering if the improvements on cross-modal retrieval may arise simply from using a larger model, and if the model really generalizes to the sister perturbation matching task.
- The clarity of the paper needs further improvement:
  - In Lines 46-61 the authors introduce too many challenges (at least 5), and in lines 62-74 I don't find a clear correspondence of how the proposed approach addresses each of the problems.
  - In Section 3.1, it took me some effort to understand why pooling profiles address both false negative and false positive problems.
  - The CWCL objective in Equ. (5) seems problematic: $exp(<p_i\cdot q_i>/\tau)\rightarrow exp(<p_i\cdot q_j>/\tau)$. Further explanations on how CWCL resolves false negatives are recommended.

---

> ### Author Rebuttal · Authors · 2025-07-31
>
> We thank the reviewer for carefully examining our work and providing their feedback. We’re thrilled to hear that you found the motivation of our work to be “clear and closely related to real-world scientific applications” and that you recognize our results “show significant improvements over prior works”. In the space below we respond to the reviewer’s specific concerns. **Due to space constraints, our response to the reviewer's questions/minor concerns can be found in the response box for CsPQ. Please let us know if we can further clarify any of these points.**
>
> # **Weaknesses/major concerns:**
>
> ### **Re: How do we know that high-content screening (HCS) data has false positives and false negatives?**
>
> We thank the reviewer for raising this and for allowing us to clarify these points. First, we emphasize that: **in general, cellular perturbation screens - including high-content screening (HCS) datasets - are well-known to be substantially impacted by false positive and false negative pairs.**
>
> **False positive pairs often arise due to technical limitations with cellular perturbation technologies.** For instance, in genetic perturbation screens, CRISPR guide RNAs are used to perturb specific genes. However, due to off-target guide effects (i.e., guide has the wrong effect) [1] or low guide efficiency (i.e., guide does not have an effect) [2], guide RNAs often fail to induce the intended gene knockdown. Indeed, previous work [3] has estimated that only 5%-10% of guide RNAs have their intended effect >50% of the time. depending on the specific guide. Similarly, in chemical screens, exposure effects on cells may vary depending on cells’ positions on experimental plates, leading some cells to be weakly or not at all affected by the intended perturbation. To mitigate the impact of false positive pairs, we thus performed per-perturbation pooling of cellular profiles using the ABMIL pooling operator, which adaptively attends to the samples most representative of a perturbation.
>
> In response to your comment, to better understand the impact of accounting for false positives by focusing on the most representative samples within a perturbation, we assessed the impact of removing the most highly attended samples by ABMIL (i.e., those most likely to reflect effective perturbation) for the Bray et al. cross-modality retrieval tasks. We found that doing so led to significant performance degradations (see table below), indicating that, for each perturbation, only a relatively small number of cells may exhibit clear perturbation effects.
>
> | Removal # of top contributing instances | Chem-to-Image (%) R@1 | Chem-to-Image (%) R@5 | Chem-to-Image (%) R@10 |Image-to-Chem (%) R@1 | Image-to-Chem (%) R@5 | Image-to-Chem (%) R@10 |
> | - | - | - | - | - | - | - |
> | 0  | 1.32 | 4.59 | 7.51 | 1.18 | 4.82 | 7.56 |
> | 1  | 1.08 | 4.60 | 7.13 | 1.04 | 4.58 | 7.61 |
> | 2  | 1.13 | 4.25 | 6.38 | 0.99 | 4.34 | 6.47 |
> | 3 | 1.13 | 4.20 | 5.91 | 0.89 | 4.06 | 6.09 |
> | 4 | 0.04 | 2.50 | 5.01 | 0.09 | 3.76 | 5.34 |
> | 5 | 0.0 | 1.46 | 3.68 | 0. | 2.88 | 4.34 |
>
> **False negative pairs arise due to the fact that many cellular images within an HCS dataset share the exact same perturbation label.** For example, in the RxRx3-core dataset considered in our manuscript, each perturbation is associated with ~92 cellular images. Analogous to findings with CLIP-like models applied in other domains [Wang et al., Zhao et al., Srinivasa et al.], this fact can introduce another source of instability during model training, as a model trained with the standard CLIP loss may exhibit pathological behavior when forced to push apart cells with highly similar morphological profiles and identical perturbation labels. Indeed, as demonstrated in Table 2 in the manuscript, accounting for false negatives by training on per-perturbation pooled profiles (as opposed to individual cells) resulted in substantially improved performance.
>
> Moreover, as shown in previous works introducing new HCS datasets [4, 5], groups of related perturbations (e.g. gene knockouts targeting genes in the same higher-level gene pathway), may have near-identical morphological effects. This fact further exacerbates the issue of false negatives, as cells with semantically similar image profiles yet distinct perturbations may be spuriously pushed apart in the embedding space. To mitigate this effect, we employed the CWCL loss [6], which assigns non‐zero weights, $w_{i,j}$ in eq (5), to semantically similar negative pairs, within CellCLIP rather than the standard contrastive loss, which treats all false negatives as unrelated (zero weight). As shown in our gene-gene relationship recovery results in Table S.4, using CWCL resulted in better preservation of known relationships between perturbations (recall = 0.595) compared to the standard contrastive loss (recall = 0.553).
>
> ### **Re: Is the superior performance of natural language for perturbation encoding just due to increased model size?**
>
> We thank the reviewer for raising this question. In response, we conducted an additional set of experiments (see table below) using BERT models with 64M and 11.3M from [7] as the text encoder for CellCLIP for the Bray et al. cross-modal retrieval task. Note that the MPNN+ graph-neural-network based encoder used in our MolPhenix baseline had ~10 M parameters. While we observed a slight decrease in cross-modal retrieval performance compared to the full BERT model, we found that even the 11.3M parameter BERT model substantially outperformed the MPNN+ model.
>
> These results suggest that the improvements we observe in cross‑modal retrieval are not simply artifacts due to a larger model size, but rather stem from the use of our text-based encoding strategy.
>
> | # BERT params | Chem-to-Image (%) R@1 | Chem-to-Image (%) R@5 | Chem-to-Image (%) R@10 |Image-to-Chem (%) R@1 | Image-to-Chem (%) R@5 | Image-to-Chem (%) R@10 |
> | - | - | - | - | - | - | - |
> | 110M (ours) | 1.18 (0.20) | 4.49 (0.06) | 7.37 (0.20) | 1.25 (0.10) | 4.82 (0.10) | 7.39 (0.23) |
> | 64M | 1.53 (0.17) | 4.40 ( 0.21) | 6.90 (0.18) | 1.41 (0.17) | 4.67 (0.24) | 6.98 (0.14) |
> | 11.3M | 1.13 (0.14) | 4.04 (0.31) | 6.28 (0.19) | 1.28 (0.17) | 4.11 (0.24) | 6.20 (0.19) |
>
> ### **Re: Improvements to clarity.**
>
> We thank the reviewer for flagging these clarity issues. We provide our responses to clarify the specific points raised by the reviewer below. In the camera-ready final manuscript we will use the additional 10th page to clarify these points.
>
> * #### **Re: How do the components of our approach (lines 62-74) address the challenges identified in lines 46-61?**
>     To clarify, the challenges identified in lines 46-61 are:
>     \
>     **a.** Limited availability of image encoding models for Cell Painting microscopy data
>
>     **b.** Microscopy image channels each capturing semantically distinct information
>
>     **c.** False negative pairs due to Cell Painting data being “many-to-one” (i.e., many cells share the same perturbation label)
>
>     **d.** False positive pairs due to variable perturbation efficacy
>
>     **e.** Difficulty encoding multiple classes of perturbations with a single perturbation encoder
>     \
>     Our approach addresses these challenges as follows:
>     \
>     **1)** CrossChannelFormer allows us to reuse openly available state-of-the-art vision embedding models trained on natural images for Cell Painting data, while accounting for the distinct semantic meaning encoded in different Cell Painting image channels (challenges **a** and **b**).
>     **2)** Pooling cell image profiles mitigates the impact of both false negative pairs and false positive pairs (challenges **c** and **d**).
>     **3)** Using a natural-language-based perturbation encoder allows us to encode multiple classes of perturbations with a single model (challenge **e**).
>     \
>     As demonstrated by the ablation experiments in our manuscript, **each of these modifications** led to improved performance on downstream tasks.
>
> * #### **Re: Why does pooling address both false negatives and false positives?**
>     Thank you for pointing out this area of confusion. To clarify:
>     \
>     Pooling addresses the issue of false negatives by preventing cells with the same perturbation label from spuriously being considered as negative pairs. If these cells were incorrectly considered as negative pairs, the model may be forced to separate embeddings of cells with highly similar morphological profiles, which could lead to instability during training. By instead operating on pooled profiles, negative pairs will always correspond to distinct perturbations, potentially resulting in better performance.
>     \
>     At the same time, pooling addresses the issue of false positives by preventing representations of images assigned the same perturbation label, but which had divergent outcomes (e.g. one cell had a successful perturbation, while the other did not due to CRISPR guide efficiency issues) from being erroneously pushed together. By instead pooling morphological profiles, we may sidestep such issues stemming from variable perturbation efficacy, resulting in improved model performance.
>
> * #### **Re: “The CWCL objective in Equ. (5) seems problematic”**
>
>     This was indeed a typo. Thank you for your attention to detail, and we will fix this in the final manuscript.
>
> **References:**
> [1] https://www.nature.com/articles/nmeth.4293
> [2] https://www.nature.com/articles/nbt.3026
> [3] https://www.nature.com/articles/s41586-018-0686-x
> [4] https://www.nature.com/articles/s41592-024-02241-6
> [5] https://www.biorxiv.org/content/10.1101/2023.08.13.553051v1.full
> [6] https://arxiv.org/abs/2309.14580
> [7] https://arxiv.org/pdf/1908.08962
>
> ---
> ## **Please see the rebuttal box for reviewer CsPQ for our responses to questions/minor concerns.**

---

> ### Comment · Reviewer_Hzy8 · 2025-08-01
>
> I appreciate the authors' efforts in providing a comprehensive response and additional experimental results. My concerns have been satisfactorily addressed, and accordingly, I have revised my scores and now recommend acceptance.
>
> As an additional (non-mandatory) point of curiosity that does not affect my final decision:
>
> - The authors highlight that SMILES strings are the most influential component in the text prompts. In light of this, I wonder whether KV-PLM [1]—a BERT-based model that is continually pre-trained on biomedical texts and SMILES strings—might serve as a more suitable choice for the text encoder.
>
> Refs.
>
> [1] A deep-learning system bridging molecule structure and biomedical text with comprehension comparable to human professionals.

---

> > ### Author Response · Authors · 2025-08-05
> >
> > Based on your recommendation, we further evaluated retrieval performance using KV-PLM as the text encoder. The results are summarized below. Interestingly, while KV-PLM outperforms BioMedBERT, it still underperforms relative to our original choice.
> >
> > | Perturb. Encoder | Chem-to-Image (%) R@1 | Chem-to-Image (%) R@5| Chem-to-Image (%) R@10 | Image-to-Chem (%) R@1 | Image-to-Chem (%) R@5 | Image-to-Chem (%) R@10 |
> > |-|-|-|-|-|-|-|
> > | BiomedBERT | 1.46 (0.10) | 4.11 (0.13) | 6.19 (0.26) | 1.51 (0.09) | 4.34 (0.13) | 6.57 (0.19) |
> > | KV-PLM | 1.29 (0.19) | 4.41(0.31) | 6.92(0.32) | 1.41(0.16) | 4.57(0.3) | 6.81(0.21) |
> > | BERT |1.18 (0.20) | 4.49 (0.06) | 7.37 (0.20) | 1.25 (0.10) | 4.82 (0.10) | 7.39 (0.23) |
> >
> > However, we would like to emphasize that CellCLIP is a general and flexible framework, designed to allow users to represent perturbations using diverse metadata tailored to their specific needs.

---

> > > ### Comment · Reviewer_Hzy8 · 2025-08-06
> > >
> > > Thank you for providing the additional experimental results. The findings are interesting, and I concur that the generality and flexibility of the proposed framework have the potential to make a significant impact on the community. I would like to reiterate my support for the acceptance of this paper.

---

### Official Review · Reviewer_CsPQ · 2025-06-22

**Clarity:** 3
**Significance:** 3
**Originality:** 3
**Rating:** 5
**Confidence:** 2

**Summary:**

This paper introduces CellCLIP, a cross-modal contrastive learning framework for HCS data. CellCLIP leverages pre-trained image encoders coupled with a novel channel encoding scheme to better capture relationships between different microscopy channels in image embeddings, along with natural language encoders for representing perturbations. Besides the good performance, this approach does not require to pre-train a model which comes with significant advantages over many similar approaches (e.g. Kraus et. al 24). Indeed, pre-training is data and compute intense.

I think that the paper is relevant and very well executed. I recommend an acceptance.

Disclaimer: Due to the very large amount of papers I got assigned to review I am forced to keep my reviews high-level and could not dive into technical details.

**Questions:**

None

**Ethical Concerns:**

["NO or VERY MINOR ethics concerns only"]

**Final Justification:**

I keep my score and recommend the acceptance of the paper.

**Paper Formatting Concerns:**

-

**Quality:**

4

**Strengths And Weaknesses:**

Strengths
- Careful system design with well-motivated architectural choices (e.g., pooling, channel reasoning, text encodings).
- Comprehensive experimental validation across retrieval and biological interpretability tasks.
- Strong quantitative improvements over baselines. This is particularly remarkable as some of the baselines have been produced using very large computational resources.
- Efficient implementation with substantial speed-ups over previous work.

Weaknesses:
None

---

> ### Author Rebuttal · Authors · 2025-07-31
>
> Dear reviewer CsPQ,
>
> Thank you for the kind words and for voting for acceptance. We’re thrilled to hear that you believe our approach “comes with significant advantages over many similar approaches”, and that our paper “is relevant and very well executed”.
>
> If there is any other information we can provide/experiments we could run during the discussion period that would allow you to further raise your score, please let us know.
>
> Thanks again,
>
> **Due to space constraints, we have dedicated the remainder of the space to responding to Reviewer Hzy8's questions/minor concerns.**
>
> ---
>
> ## **Questions/minor concerns from Reviewer Hzy8:**
>
> ### **Re: How do the different components of the prompt affect model performance?**
>
> We thank the reviewer for raising this question. In response, we performed an ablation study where we retrained and re-evaluated CellCLIP after removing different components of our prompt (e.g. cell type, drug name, SMILES string, etc.) on the cross-modal retrieval task for the Bray et al. dataset. We found that removing the SMILES string led to a large degradation in performance, and that removing cell type information led to a smaller, but consistent, degradation in performance. On the other hand, removing drug name did not negatively affect the model’s performance, likely due to any relevant information already being captured in the richer SMILES representation.
>
> | Ablation | Chem-to-Image (%) R@1 | Chem-to-Image (%) R@5 | Chem-to-Image (%) R@10 | Image-to-Chem (%) R@1 | Image-to-Chem (%) R@5 | Image-to-Chem (%) R@10 |
> |-|-|-|-|-|-|-|
> | Original | 1.18 (0.20) | 4.49 (0.06) | 7.37 (0.20) | 1.25 (0.10) | 4.82 (0.10) | 7.39 (0.23) |
> | No cell info | 1.11 (0.13) | 4.02 (0.25) | 6.59 (0.33) | 1.10 (0.23) | 4.07 (0.29) | 6.36 (0.32) |
> | No drug name  | 1.47 (0.29) | 4.77 (0.20) | 7.37 (0.29) | 1.61 (0.26) | 4.89 (0.19) | 7.21 (0.27) |
> | No SMILES | 0.13 (0.05) | 0.68 (0.19) | 1.26 (0.14) | 0.14 (0.07) | 0.63 (0.17) | 1.24 (0.13) |
>
> ### **Re: Why did we specifically use BERT for text encoding, and why not use a domain-specific model (e.g. BioMedBERT)?**
> We decided to use BERT (e.g. as opposed to the encoder from the original CLIP model) because we required an encoder with sufficient **input context length** to enable sufficiently rich and detailed textual inputs. For example, the text encoder from the original CLIP model is limited to a 77‑token context window. This limited window is insufficient for representing SMILES strings (the average length of a SMILES string is 98 characters in the Bray et al. dataset) together with cell information and other metadata. On the other hand, BERT supports up to 512 tokens. Moreover, several works [1, 2, 3] have shown promising performance with BERT-based text encoder for vision-language alignment tasks.
> In response to the reviewer’s second question regarding the performance of domain-specific models, **we conducted an additional set of experiments using the domain-specific BioMedBERT as our text encoder** for the Bray et al. cross-modality retrieval tasks. We found (see table below) that the performance of BioMedBERT was largely inferior to that of the standard BERT model across various choices of vision encoder. This may be because many of the compounds in Bray et al. are experimental and thus not represented in BioMedBERT's training corpus, which consists of PubMed and PubMed Central articles.
> | Perturb. Encoder | Chem-to-Image (%) R@1 | Chem-to-Image (%) R@5| Chem-to-Image (%) R@10 | Image-to-Chem (%) R@1 | Image-to-Chem (%) R@5 | Image-to-Chem (%) R@10 |
> |-|-|-|-|-|-|-|
> | BiomedBERT | 1.46 (0.1)| 4.11 (0.13) | 6.19 (0.26) | 1.51 (0.09) | 4.34 (0.13) | 6.57 (0.19) |
> | BERT | 1.18 (0.20) | 4.49 (0.06) | 7.37 (0.20) | 1.25 (0.10) | 4.82 (0.10) | 7.39 (0.23) |
>
> ### **Re: Do we need so many layers for CrossChannelFormer?**
>
> Thank you for raising this question. In response, we performed an additional set of experiments investigating the impact of the number of layers in CrossChannelFormer on performance for the Bray et al. cross-modal retrieval task. Although the number of tokens (i.e., wavelength channels) in the HCS data is fewer than 10, we found that increasing the number of layers in CrossChannelFormer is necessary for achieving strong model performance.
>
> | # Layers in CrossChannelFormer | Chem-to-Image (%) R@1 | Chem-to-Image (%) R@5| Chem-to-Image (%) R@10 | Image-to-Chem (%) R@1 | Image-to-Chem (%) R@5 | Image-to-Chem (%) R@10 |
> |-|-|-|-|-|-|-|
> | 12-layer | 1.18 (0.20) | 4.49 (0.06) | 7.37 (0.20) | 1.25 (0.10) | 4.82 (0.10) | 7.39 (0.23) |
> | 6-layer | 0.79 (0.09) | 3.00 (0.27) | 5.09 (0.26) | 0.77 (0.19) | 2.94 (0.30) | 4.66 (0.34) |
> | 3-layer | 0.59 (0.08) | 2.27 (0.43) | 3.88 (0.58) | 0.58 (0.21) | 2.24 (0.6) | 3.82 (0.91) |
>
>
> ### **Re: Do we see the benefits of transfer learning when pretraining on Bray et al. and fine-tuning on CP-JUMP1?**
>
> We thank the author for raising this question. In the initial iterations of our CP-JUMP1 experiments we attempted to train CellCLIP directly on CP-JUMP1 (i.e., without a pretraining step). However, we found that doing so resulted in extremely poor performance, with CellCLIP’s mean average precision (mAP) being close to zero. We believe this poor performance was due to the relatively small number of perturbations in this dataset (811 unique perturbations, with about ~500 left for training after performing a train-val-test split).
>
> In contrast, when we first pretrained the model on the larger Bray et al. dataset and then fine-tuned on CP-JUMP1, we observed substantial improvements in downstream performance across both CellCLIP and all baseline models. **These results show clear benefits of transfer learning in this setting.**
>
> **References:**
> [1] "BLIP: Bootstrapping Language-Image Pre-training for Unified Vision-Language Understanding and Generation" https://arxiv.org/abs/2201.12086
> [2] "BLIP-2: Bootstrapping Language-Image Pre-training with Frozen Image Encoders and Large Language Models" https://arxiv.org/abs/2301.12597
> [3] "BiomedCLIP: a multimodal biomedical foundation model pretrained from fifteen million scientific image-text pairs" https://arxiv.org/abs/2303.00915

---

### Official Review · Reviewer_8uCR · 2025-07-03

**Clarity:** 4
**Significance:** 3
**Originality:** 3
**Rating:** 4
**Confidence:** 3

**Summary:**

This paper introduces a new framework to predict the perturbation effects assessed by high-content screening. The performance is assessed through the application of their method, CellCLIP, to several data sets.

**Questions:**

Can the authors evaluate the stability of their perturbation embeddings? Also, is there a way to evaluate the performance across studies?

**Ethical Concerns:**

["NO or VERY MINOR ethics concerns only"]

**Final Justification:**

Keep my score

**Limitations:**

The limitations were discussed.

**Paper Formatting Concerns:**

No concerns.

**Quality:**

3

**Strengths And Weaknesses:**

Strengths: There is a clear need to develop effective models that predict the effects of various types of perturbations using data accumulated in the literature. The authors have proposed several approaches to address the limitations of existing methods, including the one-to-many mapping of images to perturbations and the need to integrate different modes of perturbations. The performance suggests the advantages of their strategy.

Weaknesses: The embedding for perturbation based on prompt engineering may introduce uncertainties and variations across users/runs. It is also challenging to assess the generalizability of the proposed methods across different datasets. Therefore, we have several questions for the authors.

1. To ensure that the authors optimize choices of prompts, we believe that the other should consider different methods of text description. For example, we may use gene/molecular information from NCBI and Drugbank as descriptions, and text encoders from OpenAI or Gemini as perturbation encoders. These choices should be considered in the benchmarking analysis.

2. The authors support design of their proposed multi-cell encoder. How to select the best number of images used in each encoder, and could we just use the mean avenge are the simple aggregator?

3. The authors should include hyper parameter information and justify the comparison is fair and other methods are also optimized in the benchmarking set.

4. Since we have large-scale cell painting data, is it possible to pre-train an image encoder rather than using the DINOv2 for normal image set? Will it be helpful for having a better performance?

---

> ### Author Rebuttal · Authors · 2025-07-31
>
> We thank the reviewer for carefully examining our work and providing their feedback. We’re happy to hear that the reviewer recognizes the “clear need to develop effective models” in this area, and we’re thrilled that the reviewer found that our results “suggests the advantages of [our] strategy”. Below, we respond to the reviewer’s concerns. Please let us know if we can further clarify these points.
>
> ### **Re: Optimizing the choice of prompt**
>
> We thank the reviewer for raising this important point. While we acknowledge that CellCLIP’s performance may vary with prompt format, it is designed as a general, flexible framework that allows users to represent perturbations using diverse metadata tailored to their needs; we note that the prompt template used in our paper is just one example. Although the current prompting strategy consistently achieves state-of-the-art performance across all evaluated datasets, we agree that prompt engineering could offer room for further improvement. In response to the reviewer’s suggestion, we explored alternative prompting strategies. These provided insights into component-level contributions, but none of them outperformed the original. Results are detailed below:
>
> First, we performed an ablation study where we re-trained and re-evaluated CellCLIP after removing different components of our original prompt template (e.g. cell type, drug name, SMILES string, etc.) on the cross-modal retrieval task for the Bray et al. dataset. We found that removing the SMILES string led to a large degradation in performance, and that removing cell type information led to a smaller degradation. On the other hand, removing drug names did not negatively affect the model’s performance, likely due to relevant information already being captured in the richer SMILES representation.
>
> Second, following the reviewer’s advice, we experimented with incorporating descriptions from an external source into the prompt template for CellCLIP. Specifically, we added DrugBank descriptions for any compounds that could be found in the database (correspondsing to approximately 10% of the compounds in the Bray et al. (2017) dataset). We note that the match rate was low, as many compounds in this dataset are small experimental molecules. After incorporating available DrugBank descriptions into the prompts, we found no improvement in performance compared to our original template. While it is possible that retrieval performance could benefit from more comprehensive metadata, this was not feasible given the limited annotation coverage in DrugBank. Nonetheless, these results suggest that our original prompting strategy is already highly effective.
>
> | Ablation  | Chem-to-Image (%) R@1 |  R@5 |R@10 | Image-to-Chem (%) R@1 | R@5 |  R@10 |
> |-|-|-|-|-|-|-|
> | Original | 1.18 (0.20) | 4.49 (0.06) | 7.37 (0.20) | 1.25 (0.10) | 4.82 (0.10) | 7.39 (0.23) |
> | No cell info | 1.11 (0.13) | 4.02 (0.25) | 6.59 (0.33) | 1.10 (0.23) | 4.07 (0.29) | 6.36 (0.32) |
> | No drug name | 1.47 (0.29) | 4.77 (0.20) | 7.37 (0.29) | 1.61 (0.26) | 4.89 (0.19) | 7.21 (0.27) |
> | No SMILES | 0.13 (0.05) | 0.68 (0.19) | 1.26 (0.14) | 0.14 (0.07) | 0.63 (0.17)  | 1.24 (0.13) |
> | With DrugBank description | 1.13 (0.16) | 4.26 (0.43) | 6.94 (0.37) | 1.16 (0.17)  | 4.34 (0.36) | 6.71 (0.26) |
>
> ### **Re: Experiments with different text encoders (e.g. from OpenAI) for the perturbation encoder**
>
> We thank the reviewer for this suggestion. In response, we conducted an additional set of experiments where we used OpenAI’s `text-embeddings-3-small` model with our prompt template to generate perturbation embeddings, followed by a 4-layer MLP as the perturbation encoder for CellCLIP on the Bray et al. cross-modal retrieval tasks. We chose this embedding model specifically because embeddings from OpenAI’s decoder-only models, such as ChatGPT, are not accessible to users.  We found that this resulted in substantially worse performance compared to our original choice of BERT. This suggests that general-purpose embeddings, even from powerful embedding models, are insufficient for domain-specific tasks without fine-tuning of the base text encoder.
>
> |Perturbation Encoder| Chem-to-Image (%) R@1 |R@5|R@10| Image-to-Chem (%) R@1 |R@5 | R@10 |
> |-|-|-|-|-|-|-|
> | BERT | 1.18 (0.20) | 4.49 (0.06) | 7.37 (0.20) | 1.25 (0.10) | 4.82 (0.10) | 7.39 (0.23) |
> | OpenAI `text-embeddings-3-small` with MLP | 0.32 (0.12) | 1.72 (0.28) | 2.80 (0.19) | 0.09 (0.11) | 0.5 (0.05) | 0.89 (0.11) |
>
> ### **Re: How to select the best number of images used in each encoder?**
>
> For all experiments presented in our manuscript, when pooling images within a perturbation we always used **all** available images for that perturbation. As shown in our submission, we found that this strategy consistently yielded strong performance compared to baseline methods.
>
> ### **Re: Could we just use the mean or other simpler aggregators?**
>
> As part of our initial submission we included results from benchmarking the impact of different choices of aggregation operator on CellCLIP’s performance for the cross-modal retrieval tasks for the Bray et al. dataset. We found (Table S.2) that attention-based pooling led to superior performance compared to simpler pooling operations like the mean or median.
>
> ### **Re: Hyperparameter tuning for CellCLIP and baselines.**
>
> The results reported in our manuscript correspond to the best‑performing models selected through extensive hyperparameter searches for both CellCLIP and baseline models. The details of these sweeps can be found in Appendix E.2, which we reproduce below for the reviewer's convenience.
>
> * Benchmarking results presented in Table 1: Here the best performing CellCLIP and baseline models were selected via a grid search over the following hyperparameter ranges:
>     * Learning rate: {7 × 10⁻⁵, 5 × 10⁻⁵, 1 × 10⁻⁵, 1 × 10⁻⁴, 2 × 10⁻⁴, 3 × 10⁻⁴, 5 × 10⁻⁴, 1 × 10⁻³, 1 × 10⁻²}
>     * Warmup steps: {200, 500, 800, 1000, 1500}
>     * Number of cosine restarts: {1, 2, 3, 4}
>     * Temperature τ: {14.3, 20, 30}
> * CellCLIP ablation results (Table 2): For each row in the table we performed a hyperparameter sweep over:
>     * Learning rate: {0.0005, 0.001, 0.005}
>     * Cosine annealing restart cycles: {5, 6, 7, 8, 9, 10}
>     * Temperature τ: {14.3, 20, 30}
>     * Epochs: {50, 60, 70}
> * Sister perturbation matching & replicate detection results (Table 3): Here we swept over the following hyperparameter ranges for all multimodal models. (For both of the unimodal CA‑MAE and OpenPhenom‑S/16 models specifically, as these models’ training data already included CP-JUMP1, we did not perform any additional training steps with these models):
>     * Learning rate: {1 × 10⁻⁵, 1 × 10⁻⁴, 2 × 10⁻⁴, 3 × 10⁻⁴, 5 × 10⁻⁴, 1 × 10⁻³, 1 × 10⁻²}
>     * Epochs: {5, 10, 20, 30, 40, 50, 60, 70}
>     * Batch size: {128, 256, 512}
>     * Warmup steps: {500, 800, 1000}
>
> * Zero-shot gene-gene relationship recovery results (Table 3): Because this task involved zero‑shot retrieval, for all multimodal models we reused the models pretrained on Bray 2017 with no additional fine tuning. For unimodal models we did not perform any additional training steps.
>
> ### **Re: Is it possible to pre-train an image encoder on Cell Painting data rather than use an encoder pretrained on natural images?**
>
> In our original submission (Appendix  A.1), we also experimented with using a specialized image encoder model pretrained on Cell Painting data as the vision encoder for CellCLIP. In particular, we used OpenPhenom [1], the current state-of-the-art openly available pretrained Cell Painting image encoder. However, we found that using OpenPhenom resulted in consistently worse performance compared to natural image encoders. Indeed, OpenPhenom (178.05 M parameters) was even outperformed by much smaller natural image encoders (e.g. DINOv2 small with 22.06 M parameters). This result can be found in Table S.1 in our submission, which we reproduce below for the reviewer’s convenience.
>
> We believe that this result may be due to the relatively small amount of publicly available Cell Painting data available for training compared to natural images. For example, OpenPhenom was trained on ~3 M Cell Painting images collected from current publicly available datasets. This is far smaller than the typical size of datasets used to train natural image foundation models (e.g. ~142 M images for DINOV2 [2]), which likely reduces the potential benefits of pretraining/transfer learning.
>
> | Image Encoding Backbone | # of Params | Perturb-to-profile (%) R@1 |  R@5 |  R@10 | Profile-to-perturb (%) R@1 | R@5 |  R@10 |
> |-|-|-|-|-|-|-|-|
> | DINOV1 | 86.4 M | 0.75 ± 0.08 | 2.77 ± 0.13 | 4.74 ± 0.33 | 0.86 ± 0.16 | 2.77 ± 0.27 | 4.80 ± 0.19 |
> | DINOV2 (small) | 22.06 M | 1.19 ± 0.32 | 4.06 ± 0.27 | 6.57 ± 0.33 | 1.32 ± 0.17 | 4.34 ± 0.39 | 6.63 ± 0.19 |
> | DINOV2 (base) | 86.58 M | 1.19 ± 0.12 | 4.03 ± 0.20 | 6.24 ± 0.40 | 1.25 ± 0.12 | 3.95 ± 0.18 | 6.32 ± 0.38 |
> | DINOV2 (large) | 304.37 M | 0.97 ± 0.12 | 4.33 ± 0.43 | 6.76 ± 0.54 | 1.16 ± 0.33 | 4.23 ± 0.40 | 6.50 ± 0.47 |
> | DINOV2 (giant) | 1136.48 M   | 1.18 ± 0.20 | 4.49 ± 0.06 | 7.37 ± 0.20  | 1.25 ± 0.10 | 4.82 ± 0.10 | 7.39 ± 0.23 |
> | OpenPhenom-S/16 | 178.05 M | 0.94 ± 0.04 | 3.73 ± 0.13 | 6.22 ± 0.20 | 1.27 ± 0.14 | 4.13 ± 0.19 | 6.25 ± 0.08  |
>
> ### **Re: Can the authors evaluate the stability of their perturbation embeddings? Also, is there a way to evaluate the performance across studies?**
>
> To assess the stability of our method, all results in our experiments corresponded to trials with three random seeds. Moreover, to ensure the generality of our method our experiments contained results from multiple datasets (Bray et al., CPJUMP1, and RxRx3 Core). **Please let us know if you would like further information or if we have misunderstood your question.**
>
> **References:**
>
> 1: https://huggingface.co/recursionpharma/OpenPhenom
>
> 2: DINOv2: Learning Robust Visual Features without Supervision, https://arxiv.org/pdf/1908.08962

---

> > ### Comment · Reviewer_8uCR · 2025-08-04
> > **Thank you**
> >
> > Thank you for your responses. I will keep my score and stay in the acceptance side.

---

### Author Response · Authors · 2025-08-07

We would like to thank the reviewers for their insightful and important questions, as well as for the opportunity to clarify the contributions of our work. We also appreciate the active engagement and prompt responses throughout the rebuttal and discussion phases.

---

### Note · Authors · 2025-08-15

We thank the reviewers again for their insightful reviews and engagement during the rebuttal process.  We believe that the additional experimental results produced during the rebuttal period per the reviewers' suggestions have strengthened the arguments from our original submission, and we're thrilled that the reviewers agree, with all reviewers now recommending acceptance. Thank you again for a productive reviewing cycle.

---

### Decision · Program_Chairs · 2025-09-17

**Decision:**

Accept (poster)

**Comment:**

The paper proposes CellCLIP, a cross-modal contrastive learning framework for connecting cell painting images with textual perturbation descriptions. Reviewers praised the strong technical contributions, and comprehensive experiments demonstrating consistent gains over baselines.
Initial concerns focused on the clarity of the false positive and false negative issues, the choice of text encoders, and generalization. The authors provided detailed rebuttals with additional experiments, including ablations, comparisons with domain-specific encoders, and transfer learning analyses. These satisfactorily addressed all concerns, and reviewers updated or confirmed their positive recommendations.

With all reviewers supporting acceptance after rebuttal, this paper is recommended for acceptance.